# Time-dependent cytokine and chemokine changes in mouse cerebral cortex following a mild traumatic brain injury

David Tweedie[1†]*, Hanuma Kumar Karnati[1†], Roger Mullins[2], Chaim G Pick[3], Barry J Hoffer[4], Edward J Goetzl[5], Dimitrios Kapogiannis[2], Nigel H Greig[1]

[1]Translational Gerontology Branch, Intramural Research Program, National Institute on Aging, NIH, Baltimore, United States; [2]Laboratory of Clinical Investigation, Intramural Research Program, National Institute on Aging, NIH, Baltimore, United States; [3]Department of Anatomy and Anthropology, Sackler School of Medicine, Sylvan Adams Sports Institute, and Dr. Miriam and SheldonG. Adelson Chair and Center for the Biology of Addictive Diseases, Tel Aviv University, Tel Aviv, Israel; [4]Department of Neurosurgery, Case Western Reserve University School of Medicine, Cleveland, United States; [5]Department of Medicine, University of California Medical Center, San Francisco, San Francisco, United States

**Abstract** Traumatic brain injury (TBI) is a serious global health problem, many individuals live with TBI-related neurological dysfunction. A lack of biomarkers of TBI has impeded medication development. To identify new potential biomarkers, we time-dependently evaluated mouse brain tissue and neuronally derived plasma extracellular vesicle proteins in a mild model of TBI with parallels to concussive head injury. Mice (CD-1, 30–40 g) received a sham procedure or 30 g weight-drop and were euthanized 8, 24, 48, 72, 96 hr, 7, 14 and 30 days later. We quantified ipsilateral cortical proteins, many of which differed from sham by 8 hours post-mTBI, particularly GAS-1 and VEGF-B were increased while CXCL16 reduced, 23 proteins changed in 4 or more of the time points. Gene ontology pathways mapped from altered proteins over time related to pathological and physiological processes. Validation of proteins identified in this study may provide utility as treatment response biomarkers.

*For correspondence:
tweedieda@grc.nia.nih.gov

[†]These authors contributed equally to this work

Competing interests: The authors declare that no competing interests exist.

## Introduction

Traumatic brain injury (TBI) is a common health concern that affects thousands to millions of people in the United States and worldwide (*Thurman et al., 1999*; *Cassidy et al., 2004*; *Centers for Disease Control and Prevention, 2010*). The consequences of TBI manifest in a time-dependent manner and unfold in phases starting immediately following the traumatic event. The primary damage to the brain tissue is followed by a series of secondary molecular and cellular events that can either cause reversible or permanent cell damage resulting in cell death, tissue loss and, in time, neurodegeneration. In many cases, changes in subjects' cognition and personality induced by a mild TBI (mTBI) spontaneously resolve over time; however, this is not always true (*Thornhill et al., 2000*; *Arciniegas et al., 2005*). Some individuals are left with long-lasting deficits in cognition and executive functions, which adversely affect the patient's and their families' quality of life. mTBI is hence hugely costly to health care systems internationally and incurs a substantial national financial and quality of life burden (*Thurman et al., 1999*; *Dillahunt-Aspillaga et al., 2013*; *Qadeer et al., 2017*).

Presently, there are no FDA-approved first-choice medicines for the treatment of TBI (*Margulies and Hicks, 2009*; *Diaz-Arrastia et al., 2014*). This is a critical roadblock in the effective

clinical management of TBI. A lack of robust biomarkers to quantitatively assess TBI severity and predict its time-dependent progression with activation of secondary injury cascades, as well as its potential resolution, has proven to be a significant impediment in the development of effective treatments. Several studies have recently provided positive headway by suggesting candidate biomarkers for TBI severity and prediction of patient outcomes. Candidate biomarkers for moderate-to-severe TBI measured in serum/plasma or cerebrospinal fluid (CSF) reflect damage to various brain cell types and include neuronal-specific ubiquitin C-terminal hydrolase L1 (UCH-L1, *Korley et al., 2018*; *Mondello et al., 2011*), neuronal-specific neurofilament light (NFL, *Korley et al., 2018*; *Shahim et al., 2016*; *Olczak et al., 2018*), astrocytic-specific glial fibrillary acidic protein (GFAP, *Mondello et al., 2011*; *Olczak et al., 2018*) and oligodendrocyte-specific myelin basic protein (MBP, *Olczak et al., 2018*).

Regarding S100 calcium binding protein B (S100B), there have been mixed data on its utility in TBI; *Golden et al., 2018* and *Kövesdi et al., 2010* found it to predict poor outcomes following severe TBI, whereas *Shahim et al., 2016* determined that it was unhelpful in predicting patient outcomes (*Golden et al., 2018*; *Kövesdi et al., 2010*; *Shahim et al., 2016*). Potential biomarkers for mTBI have likewise been investigated, and plasma/serum levels of NFL, UCH-L1 and tau (a microtubule-associated protein) were observed to be potentially useful (*Joseph et al., 2018*; *Shahim et al., 2017*). Chen and co-workers observed elevations in RNA transcripts of genes implicated in Alzheimer's disease (Cyclin Dependent Kinase 2, Casein Kinase 1 Alpha 1 and Cathepsin D) in saliva from patients with mild, concussion-related TBI (*Cheng et al., 2019*). While progress has been made for identifying biomarkers for moderate-to-severe TBI, more effort needs to be applied in identifying effective biomarkers of mTBI, which comprises about 85% of all TBI events.

Our laboratories have utilized a murine closed head weight drop model of mTBI, modified from the Marmarou model (*Marmarou et al., 1994*), that mimics numerous features of the human condition (*Zohar et al., 2003*; *Milman et al., 2005*; *Tashlykov et al., 2007*; *Tweedie et al., 2007*). Several of our prior studies focused on hippocampal gene expression changes, which allowed us to identify key molecular pathways triggered by a single mTBI insult that were successfully mitigated by pharmacological treatment (*Tweedie et al., 2013a*; *Tweedie et al., 2013b*; *Tweedie et al., 2016a*; *Tweedie et al., 2016b*).

The present study follows up on this earlier research by evaluating changes in brain tissue proteins induced by a single mTBI event. Based on our prior gene array evidence highlighting the involvement of inflammatory pathways in mTBI and the work of others (*Israelsson et al., 2008*; *Israelsson et al., 2009*; *Tweedie et al., 2016b*), we quantified the levels of approximately 200 inflammation-related proteins in ipsilateral cortical tissue to define time-dependent changes. Ipsilateral cortical tissues were obtained from one sham group and mTBI animals euthanized at 8, 24, 48, 72 and 96 hr and 7, 14 and 30 days post-mTBI. We observed numerous differences in inflammatory proteins between mTBI-challenged and sham tissue samples, 23 proteins were significantly different from sham protein levels in four or more of the eight different sample time points. Notably, the proteins FASL, CXCL16, Fractalkine, IL-1 ra and MIG were significantly reduced and IL-17E, GAS-1 and VEGF-B significantly elevated by mTBI, in six to seven of the eight time points. Gene ontology pathways mapped from proteins that changed over time were associated with pathological and physiological processes and several were linked with pathways that may be beneficial or pathological, depending on which proteins were altered at the specific time points. To discover non-invasive biomarkers for mTBI, we additionally assessed the protein contents of neuronally enriched extracellular vesicles (NEVs) extracted from the plasma of a subset of mTBI-challenged mice. Correlations in protein changes between cortical tissue and CNS EV cargo proteins were explored.

## Results

### Time-dependent changes in cortical cytokine and chemokines induced by a single mTBI

Of the 200 proteins on the arrays, 161 proteins were detectable in sufficient sample numbers (i.e. n = 3 per time point) to allow for statistical analysis. Assessments of the 161 proteins by ANOVA indicated that 58 proteins on the arrays were different in mTBI-challenged mice in a minimum of one time point compared to sham mice. Comparing each mTBI time point individually to the sham values

by t-test indicated that 124 of the proteins on the arrays were different in at least one time point compared to sham. The patterns of protein changes were complex. Most proteins that were significantly changed by mTBI displayed changes in only one direction, that is elevations or reductions across time points when compared to sham protein levels. A smaller number of proteins displayed different directions in changes at different times points. For example, IGFBP-6 (lower at 24 hr and increased at 30 days); OPG (higher at 48 hr and decreased at 30 days) and Eotaxin-2 (higher at 8 hr and decreased at 72 and 96 hr). Certain proteins showed differences from sham at only one time point, examples are IL-10; IL-5; IL-12p40 and THPO; many more showed differences at several time points, for example FASL; CXCL16; TCA-3; GITR; Fractalkine; IL-1 Ra and VEGF-B. *Figures 1*, *2* and *3* illustrate statistically significant changes in protein levels as a percentage 'increase' from sham sample levels at 8 and 24 hr post-mTBI (*Figure 1*); 48 to 96 hr post-mTBI (*Figure 2*) and from 7 days to 30 days post-mTBI (*Figure 3*). *Supplementary file 1*, proteins significantly changed by a single mTBI event over time, data are percentage increase from sham, mean ± SEM and n for all significantly regulated proteins over the 30-day time period.

## Significantly altered cortical proteins at 8 and 24 hr post-mTBI

Numerous proteins were observed to be significantly changed as early as 8 hr after a single mTBI. Specifically, 22 proteins were significantly higher compared to sham; the highest were MIP-3b (+173 ± 63%, n = 7); SLAM (+168 ± 46%, n = 8), VEGF-B (+142 ± 50%, n = 8) and CRP (+104 ± 23%, n = 12). Two proteins were significantly lower compared to sham, FASL (−30 ± 8%, n = 12) and MIP-1a (−58 ± 10%, n = 9), *Figure 1*, Left - 8 hr. By 8 hr following mTBI, nine of the significantly altered proteins mapped to 10 different gene ontology pathways. The pathways were observed to potentially be deleterious to (pathological), or beneficial (physiological) to cellular processes in response to mTBI. Additional pathways had the potential to be either beneficial or deleterious to cellular processes. The name of the pathway(s) is provided with the identities of the protein(s) that were mapped to the specific pathway(s) for specific time points. Potentially, pathological pathways included Apoptosis signaling pathway (P00006, FASL), FAS signaling pathway (P00020, FASL) and Alzheimer disease-presenilin pathway (P00004, MPP-2). Potentially, physiological pathways included Angiogenesis (P00005, PDGF-AA and bFGF), PDGF signaling pathway (P00047, PDGF-AA) and EGF receptor signaling pathway (P00018, BTC). Other pathways include Inflammation mediated by chemokine and cytokine signaling pathway (P00031, MIP-1a and IFNg R1), Interferon-gamma signaling pathway (P00035, IFNg R1), Interleukin signaling pathway (P00036, gp130) and FGF signaling pathway (P00021, bFGF).

By 24 hr post-mTBI, 46 proteins were significantly lower than sham protein levels; the lowest were IL-1 ra (−75 ± 4%, n = 6); IL-12p40 (−75 ± 5%, n = 8); MCP-5 (−73 ± 5%, n = 5); HAI-1 (−68 ± 10%, n = 4). Ten proteins were higher compared to sham; the highest were CD6 (+89 ± 28%, n = 9) and E-Cadherin (+69 ± 15%, n = 8), *Figure 1*, Right - 24 hr. Fifteen of the altered proteins mapped to 12 different pathways: pathological pathways included Apoptosis signaling pathway (P00006, FASL, Granzyme B), FAS signaling pathway (P00020, FASL) and Alzheimer disease-presenilin pathway (P00004, E-Cadherin). Physiological pathways included Angiogenesis (P00005, bFGF), CCKR signaling map (P06959, E-Cadherin), Gonadotropin-releasing hormone receptor pathway (P06664, Adiponectin) and Wnt signaling pathway (P00057, E-Cadherin). Other pathways included Inflammation mediated by chemokine and cytokine signaling pathway (P00031, IFN-g, IL-6, IL-15, MIP-1a, MCP-5, IL-1b, Eotaxin, Fractalkine), Interferon-gamma signaling pathway (P00035, IFN-g), Interleukin signaling pathway (P00036, IL-6, IL-15, IL-4), FGF signaling pathway (P00021, bFGF) and Cadherin signaling pathway (P00012, E-Cadherin).

## Significantly altered cortical proteins at 48, 72 and 96 hr post-mTBI

At 48 hr following mTBI, 23 proteins were higher and 27 proteins were lower than sham (*Figure 2*, Left - 48 hr). The highest were OPN (+143 ± 56%, n = 8) and CRP (+135 ± 56%, n = 8). The lowest were MadCam-1 (−86 ± 3%, n = 3), sFRP-3 (−75 ± 3%, n = 8) and IL-1 R4 (−74 ± 5%, n = 8). Twelve different proteins significantly altered by 48 hr after mTBI mapped to 11 different functional pathways, pathological pathways were: Apoptosis signaling pathway (P00006, FASL, TNF-R2), FAS signaling pathway (P00020, FASL) and Alzheimer disease-presenilin pathway (P00004, E-Cadherin, MMP-9). Physiological pathways were: Angiogenesis (P00005, bFGF, sFRP-3), CCKR signaling map

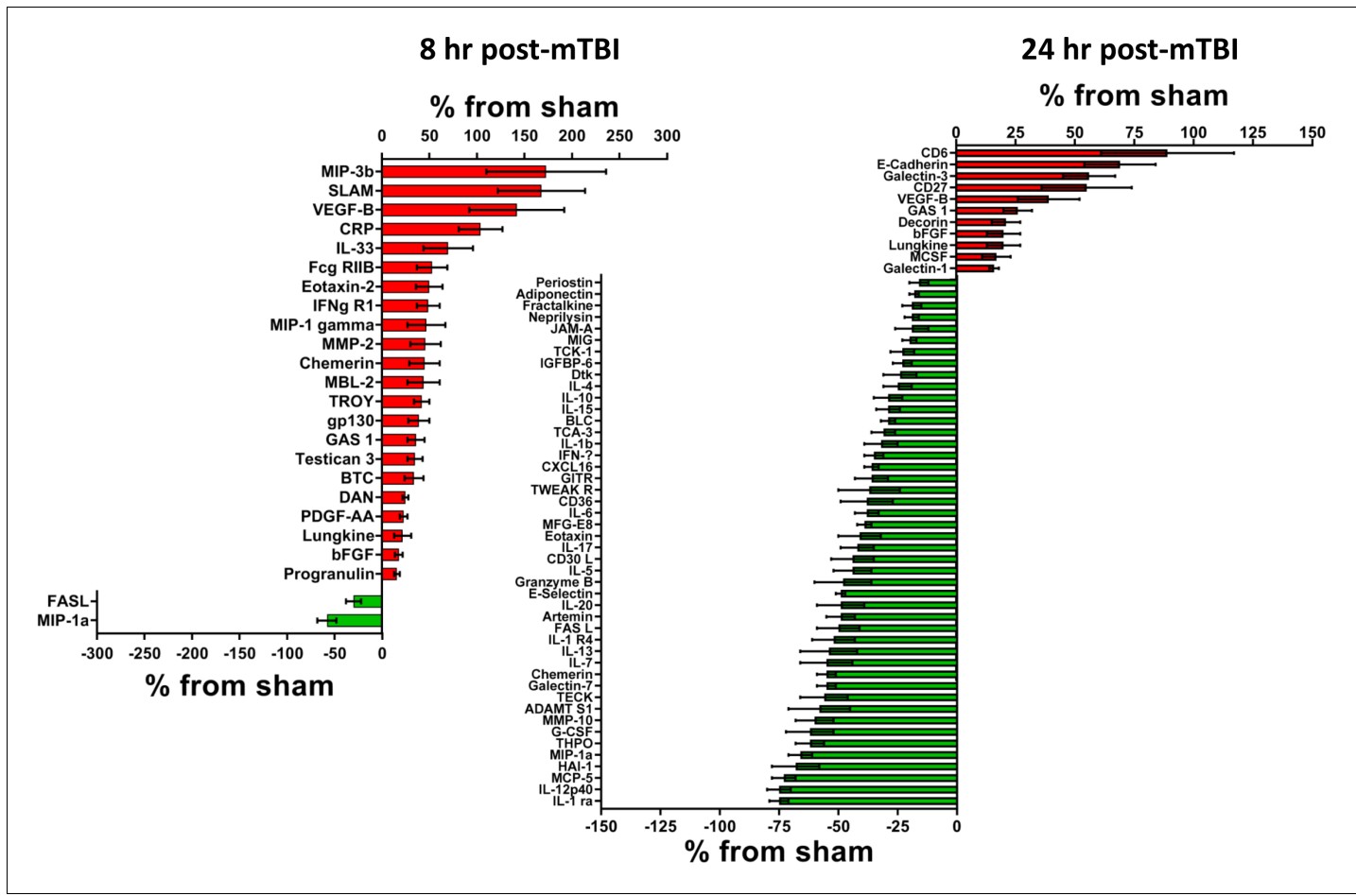

**Figure 1.** Cerebral cortex proteins significantly altered by a single mTBI event 8 and 24 hr post-injury. The percentage increase from sham levels are presented, positive values (red) are increases and negative values (green) are reductions in protein levels, compared to sham control levels. Left – At 8 hr, more proteins displayed significant increases than significant reductions in protein levels. VEGF-B; MIP-3b; SLAM and CRP were the most up-regulated proteins. FASL and MIP-1a were significantly lower than sham. Right – At 24 hr following a single mTBI event, many more proteins presented with lower rather than increased levels when compared to sham. The largest reductions were evident in IL-1 ra and IL-12p40, the largest increases in protein levels were seen for CD6 and E-Cadherin. Percentage 'increase' values are presented as the mean ± S.E.M. of n cortical samples (see *Supplementary file 1* for mean ± S.E.M. and n data). Definitions of pathways are provided in the Figure Legends, where the pathways are first mentioned in the results sections. When new pathways are described in subsequent text, the new definitions will be provided in the appropriate Figure Legend. Pathway definitions for 8 and 24 hr are: Apoptosis signaling pathway (P00006) - signal transduction pathways linked with cellular death. FAS signaling pathway (P00020) - the FAS receptor mediates apoptotic signaling initiated by interaction with surface expressed FASL on other cells, FAS-FASL apoptosis is mediated via a death domain. Alzheimer disease-presenilin pathway (P00004) - the presenilin gamma-secretase complex linked to Alzheimer's disease. Angiogenesis (P00005) - signaling pathways that have been identified as key mediators of angiogenesis. PDGF signaling pathway (P00047) - platelet-derived growth factor (PDGF) plays a critical role in cellular proliferation and development. EGF receptor signaling pathway (P00018) - signaling that mediates growth and proliferation in response to the binding of a variety of growth factor ligands. CCKR signaling map (P06959) - the classical gastrin cholecystokinin B receptor CCK-BR, its isoforms and alternative receptors, these peptides trigger signaling pathways which influence the expression of downstream genes that affect cell survival, angiogenesis and invasion. Gonadotropin-releasing hormone receptor pathway (P06664) - the GnRH receptor (GnRHR), expressed at the cell surface of the anterior pituitary gonadotrope is critical for normal secretion of gonadotropins LH and FSH, pubertal development, and reproduction. Wnt signaling pathway (P00057) - is involved in various transcriptional regulatory molecules affecting various target genes. Inflammation mediated by chemokine and cytokine signaling pathway (P00031) - this pathway illustrates chemokine-induced adhesion and migration of leukocytes resulting in the infiltration to the tissue and transcriptional activation enabling recruitment of more leukocytes. Interferon-gamma signaling pathway (P00035) - the interferon-gamma signaling pathway modulates the antiproliferative and antiviral properties of interferon-gamma. Interleukin signaling pathway (P00036) - they can mediate different biological response via activation of a combination of different signal transduction pathways. FGF signaling pathway (P00021) - the result is the activation of many downstream pathways and many cellular outcomes, including mitogenesis, differentiation, survival, apoptosis, and cell migration. Cadherin signaling pathway (P00012) - the pathway is involved in many biological processes, such as development, neurogenesis, cell adhesion, and inflammation. *Figure 1—source data 1*.

The online version of this article includes the following source data for figure 1:

**Source data 1.** Cortex tissue at 8 and 24 hrs.

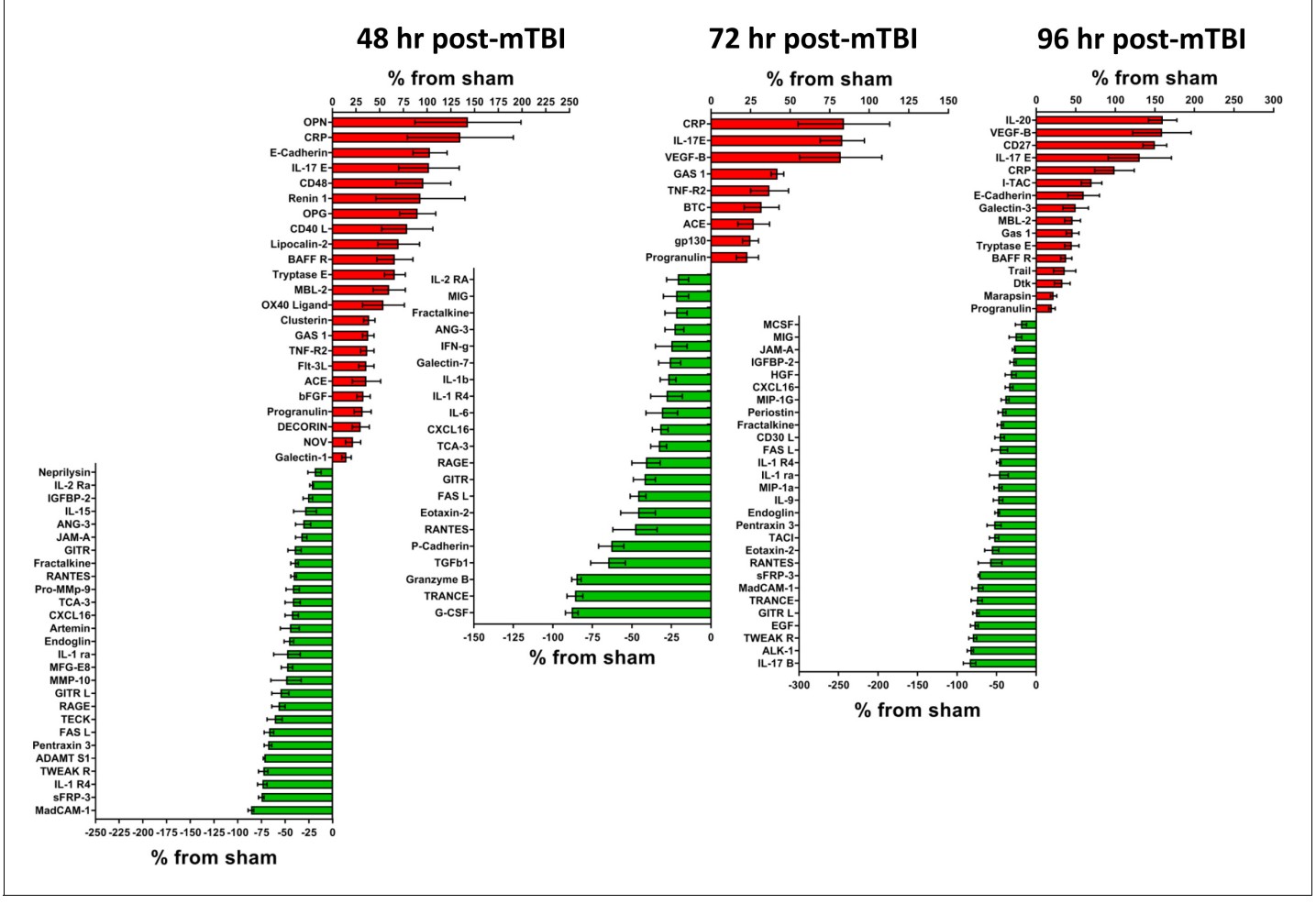

**Figure 2.** Cerebral cortex proteins significantly altered by a single mTBI event 48, 72 and 96 hr post-injury. The percentage increase from sham levels are presented, positive values (red) are increases and negative values (green) are reductions in protein levels compared to sham control. Left - 48 hr the proteins OPN, CRP, E-Cadherin and IL-17E were the most elevated compared to sham levels. MadCAM-1, sFRP-3, IL-1 R4 and TWEAK R were the most down-regulated proteins. Middle - 72 hr the largest reductions in protein levels were seen for G-CSF, TRANCE, Granzyme B and TGFb1. The largest increases in protein levels were seen for CRP, IL-17E and VEGF-B. Right - 96 hr the proteins IL-20, VEGF-B, CD27 and IL-17E were the most significantly up-regulated proteins by 96 hr. IL-17B, ALK-1, TWEAK R, EGF, GITR L, TRANCE and MadCAM-1 were the most down-regulated proteins. Percentage 'increase' values are presented as the mean ± S.E.M. of n cortical samples (see *Supplementary file 1* for mean ± S.E.M. and n data). Definitions of additional pathways observed for 48, 72 and 96 hr are: Plasminogen activating cascade (P00050) - the inactive zymogen plasminogen is converted into plasmin. TGF-beta signaling pathway (P00052) - Transforming growth factor beta signaling pathway. Other pathways have been defined elsewhere.

*Figure 2—source data 1.*

The online version of this article includes the following source data for figure 2:

**Source data 1.** Cortext issue at 48, 72 and 96hrs.

(P06959, Clusterin, MMP-9, E-Cadherin), Wnt signaling pathway (P00057, sFRP-3, E-Cadherin) and Plasminogen activating cascade (P00050, MMP-9). Other pathways were: Inflammation mediated by chemokine and cytokine signaling pathway (P00031, IL-15, RANTES, Fractalkine), Interleukin signaling pathway (P00036, IL-15, IL-2 Ra), FGF signaling pathway (P00021, bFGF) and Cadherin signaling pathway (P00012, E-Cadherin).

*Figure 2*, Middle - 72 hr, 21 proteins were lower compared with sham; the lowest were G-CSF (−88 ± 4%, n = 4), TRANCE (−86 ± 3%, n = 4) and Granzyme B (−85 ± 3%, n = 4). Nine proteins were higher, with the highest being CRP (+84 ± 29%, n = 12), IL-17E (+83 ± 14%, n = 12) and VEGF-B (+82 ± 26%, n = 9). Fourteen different proteins mapped to 11 different pathways, pathological pathways include: Apoptosis signaling pathway (P00006, TNF-R2, FASL, Granzyme B) and FAS signaling pathway (P00020, FASL) and Alzheimer disease-presenilin pathway (P00004, E-Cadherin).

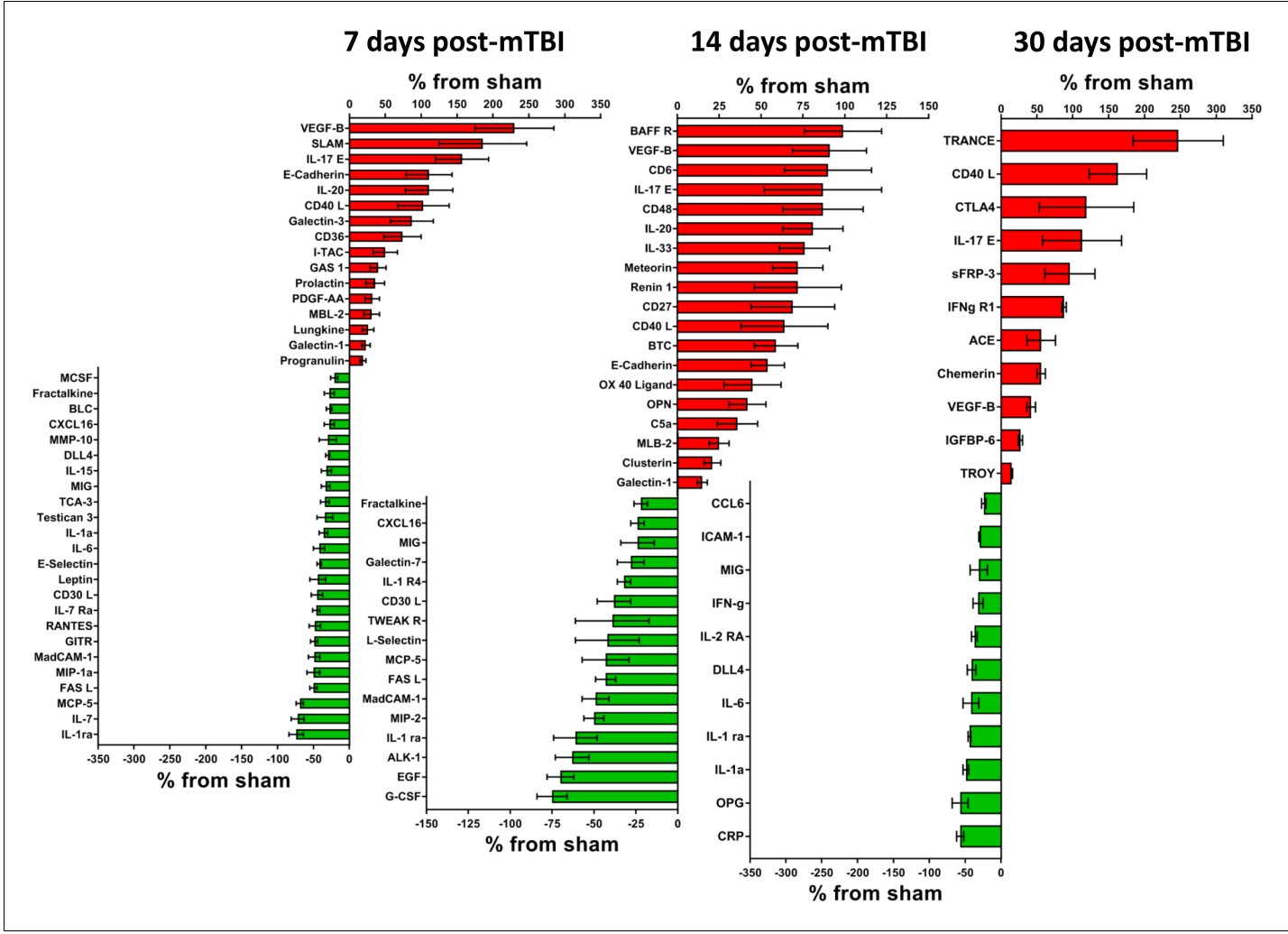

**Figure 3.** Cerebral cortex proteins significantly altered by a single mTBI event 7, 14 and 30 days post-injury. The percentage increase from sham levels are presented, positive values (red) are increases and negative values (green) are reductions in protein levels compared to sham control. Left - 7 days the proteins VEGF-B, SLAM, IL-17E, E-Cadherin and IL-20 were the most elevated compared to sham levels. IL-1 ra, IL-7, MCP-5 and FASL were the most down-regulated proteins. Middle - 14 days the largest reductions were seen for G-CSF, EGF, ALK-1 and IL-1 ra. The largest increases in protein levels were seen for BAFF R, VEGF-B, CD6 and IL-17E. Right – 30 days TRANCE, CD40 L, CTLA4, IL-17E and sFRP-3 were the most significantly up-regulated proteins by 30 days. CRO, OPG, IL-1a, IL-1 ra and IL-6 were the most down-regulated proteins. Percentage 'increase' values are presented as the mean ± S.E.M. of n cortical samples (see *Supplementary file 1* for mean ± S.E.M. and n data). An additional pathway identified for 7, 14 and 30 days is Notch signaling pathway (P00045) which is defined as follows: a transmembrane receptor that mediates local cell-cell communication and coordinates a signaling cascade. Other pathways have been defined elsewhere. *Figure 3—source data 1*.

The online version of this article includes the following source data for figure 3:

**Source data 1.** Cortext issue at 7, 14 and 30 days.

Physiological pathways include: EGF receptor signaling pathway (P00018, BTC), Gonadotropin-releasing hormone receptor pathway (P06664, TGFb1), Wnt signaling pathway (P00057, E-Cadherin) and TGF-beta signaling pathway (P00052, TGFb1). Other pathways were: Inflammation mediated by chemokine and cytokine signaling pathway (P00031, IFN-g, IL-6, Fractalkine, IL-1b, RANTES), Inter-feron-gamma signaling pathway (P00035, IFN-g), Interleukin signaling pathway (P00036, IL-6, IL-2 Ra, gp130) and Cadherin signaling (P00012, E-Cadherin).

*Figure 2*, Right - 96 hr 16 proteins were higher, with the highest being IL-20 (+160 ± 18%, n = 7), VEGF-B (+159 ± 37%, n = 8), CD27 (+150 ± 15%, n = 7) and IL-17E (+131 ± 40%, n = 8). Twenty-eight proteins were reduced, with the lowest being IL-17B (−84 ± 8%, n = 4), ALK-1 (−83 ± 4%, n = 6) and TWEAK R (−80 ± 5%, n = 8). Ten different proteins mapped to 12 different pathways:

Apoptosis signaling pathway (P00006, FASL, Trail), FAS signaling pathway (P00020, FASL) and Alzheimer disease-presenilin pathway (P00004, E-Cadherin). Physiological pathways were: Angiogenesis (P00005, sFRP-3), EGF receptor signaling pathway (P00018, EGF), CCKR signaling map (P06959, E-Cadherin), Gonadotropin-releasing hormone receptor pathway (P06664, EGF), Wnt signaling pathway (P00057, sFRP-3, E-Cadherin) and TGF-beta signaling pathway (P00052, ALK-1). Other pathways were: Inflammation mediated by chemokine and cytokine signaling pathway (P00031, Fractalkine, MIP-1a, RANTES), Interleukin signaling pathway (P00036, IL-9) and Cadherin signaling pathway (P00012, E-Cadherin).

## Significantly altered cortical proteins at days 7, 14 and 30 post-mTBI

At later time points after mTBI (7, 14 and 30 days), cortical cytokines and chemokines continued to display significant differences when compared to sham. At 7 days post-mTBI, 40 proteins were significantly different, 16 proteins were higher and 24 were lower (*Figure 3*, Left −7 days). The three highest were VEGF-B (+230 ± 57%, n = 7), SLAM (+186 ± 61%, n = 6) and IL-17E (+157 ± 37%, n = 12); the three lowest were IL-1 ra (−74 ± 10%, n = 7), IL-7 (−72 ± 9%, n = 9) and MCP-5 (−69 ± 5%, n = 5). Thirteen different proteins mapped to 12 different pathways, pathological pathways were: Apoptosis signaling pathway (P00006, FASL), FAS signaling pathway (P00020, FASL) and Alzheimer disease-presenilin pathway (P00004, E-Cadherin). Physiological pathways were: Angiogenesis (P00005, DLL4, PDGF-AA), PDGF signaling pathway (P00047, PDGF-AA), CCKR signaling map (P06959, E-Cadherin), Gonadotropin-releasing hormone receptor pathway (P06664, Prolactin), Wnt signaling pathway (P00057, E-Cadherin) and Notch signaling pathway (P00045, DLL4). Other pathways were: Inflammation mediated by chemokine and cytokine signaling pathway (P00031, IL-6, IL-15, MCP-5, MIP-1a, Fractalkine, RANTES), Interleukin signaling pathway (P00036, IL-6, IL-1a, IL-15) and Cadherin signaling pathway (P00012, E-Cadherin).

By 14 days post-mTBI, 35 proteins were significantly different, with 19 being higher and 16 lower (*Figure 3*, Middle - 14 days). The three highest were BAFF R (+99 ± 23%, n = 12), VEGF-B (+91 ± 22%, n = 10) and CD6 (+90 ± 26%, n = 8), whereas the three lowest were G-CSF (−75 ± 9%, n = 5), EGF (−70 ± 8%, n = 11) and ALK-1 (−63 ± 10%, n = 9). Seven different proteins mapped to 10 different pathways, pathological pathways were: Apoptosis signaling pathway (P00006, FASL), Fas Signaling (P00020, FASL) and Alzheimer disease-presenilin pathway (P00004, E-Cadherin). Physiological pathways were: EGF receptor signaling pathway (P00018, EGF and BTC), CCKR signaling map (P06959, Clusterin and E-Cadherin), Gonadotropin-releasing hormone receptor pathway (P06664, EGF), Wnt signaling pathway (P00057, sFRP-3, E-Cadherin) and TGF-beta signaling pathway (P00052, ALK-1). Other pathways were: Inflammation mediated by chemokine and cytokine signaling pathway (P00031, Fractalkine and MCP-5) and Cadherin signaling (P00012, E-Cadherin).

At the latest time point assessed in our study, 30 days post-mTBI, 22 proteins were significantly different compared to sham, with 11 proteins being higher and 11 lower (*Figure 3*, Right - 30 days). The three highest proteins were TRANCE (+247 ± 63%, n = 4), CD40 L (+163 ± 40%, n = 6) and CTLA4 (+119 ± 66%, n = 3); the three lowest were CRP (−57 ± 5%, n = 4), OPG (−57 ± 11%, n = 4) and IL-1a (−49 ± 4%, n = 4). At 30 days after mTBI, seven different proteins mapped to six different pathways, there were no pathological pathways, yet there were several physiological pathways: Angiogenesis (P00005, DLL4 and sFRP-3), Wnt signaling pathway (P00057, sFRP-3) and Notch signaling pathway (P00045, DLL4). Others were Inflammation mediated by chemokine and cytokine signaling pathway (P00031, IL-6, IFN-g, IFNg R1), Interferon-gamma signaling pathway (P00035, IFN-g, IFNg R1) and Interleukin signaling pathway (P00036, IL-6, IL-2 Ra, IL-1a).

## Gene ontology classifications mapped from proteins significantly altered during the study followup

Time-dependent changes in the numbers of the significantly altered proteins that mapped to the pathways at all eight time points are shown in *Figure 4A–D*. The pathways have been shown as 'Pathological - Cell Death', 'Physiological - Cell Survival' and 'Context Dependent Pathways' where the biological response will dependent on the biological function of the proteins that generated the pathway that is pro- or anti-inflammatory in activity. Provided are the names of the pathways and the numbers of altered proteins that mapped to the pathways over time. Pathological Cell Death pathways were observed from the earliest time point until 14 days following the mTBI, and the

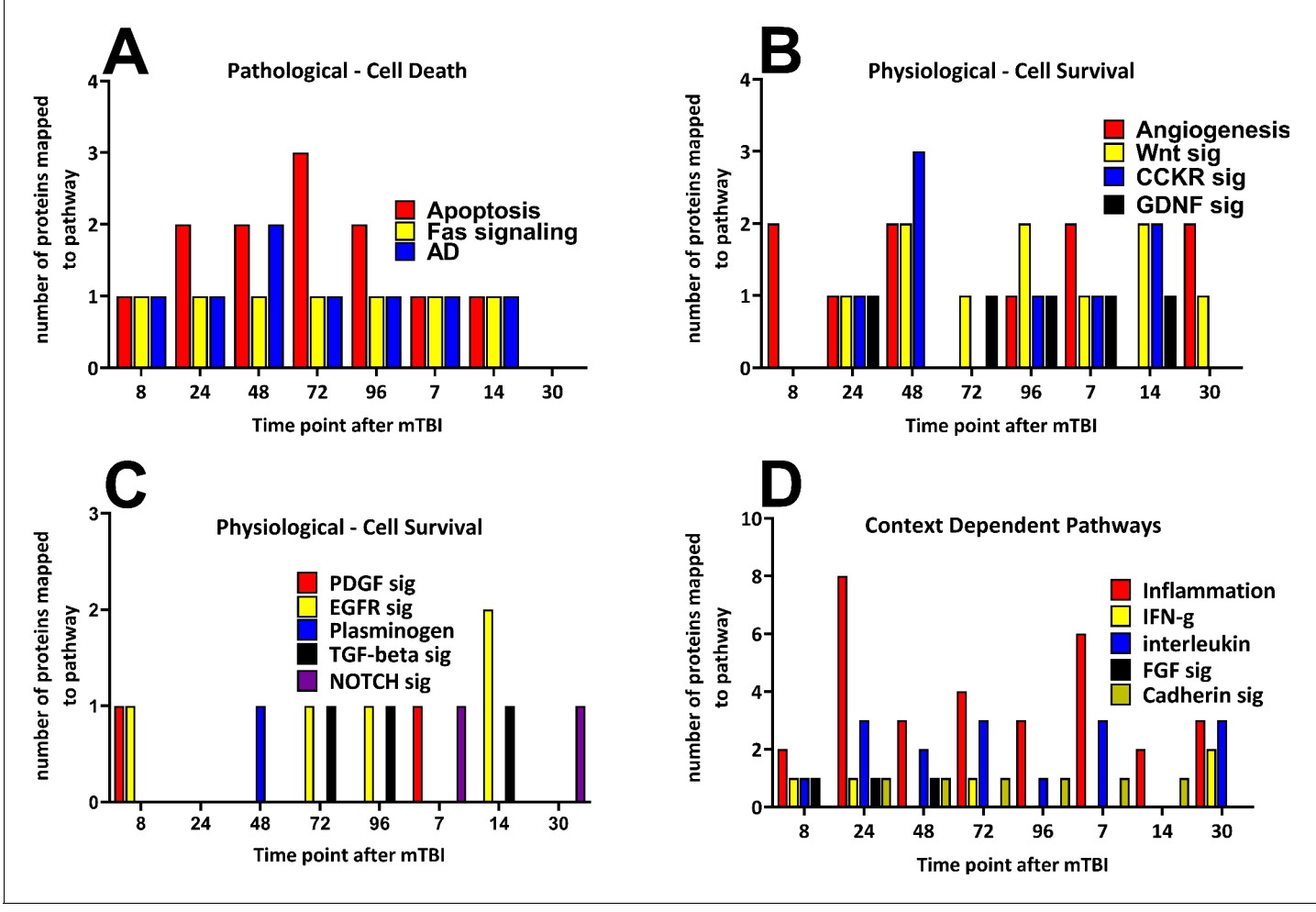

**Figure 4.** Time-dependent changes in pathways and cerebral cortical derived proteins that mapped to the pathways after a single mild TBI. Changes in protein numbers that mapped to Pathological – Cell Death (**A**), Physiological – Cell Survival (**B/C**) and Context-Dependent Pathways (**D**) are provided. Pathological pathways were observed from 8 hr up to, but not after 14 days following injury, the apoptosis pathway showed the largest number of proteins at 72 hr after injury (**A**). Physiological pathways were shown to be observed at times similar to that of Pathological pathways, yet in contrast to those, cell survival pathways were present at the 30 day time point. The numbers of proteins that mapped to the cell survival pathways were similar during the study (**B/C**). Context-dependent pathways were also observed at all times during the study, the pathways which had the largest number of proteins was the inflammation pathway (eight proteins by 24 hr and six proteins at 7 days following TBI (**D**). The pathways were derived from the identities of the proteins significantly regulated at each time point which were analyzed by the Protein ANalysis THrough Evolutionary Relationships (PANTHER) bioinformatic tool. *Figure 4—source data 1*.

The online version of this article includes the following source data for figure 4:

**Source data 1.** Pathway.

numbers of proteins that mapped to these pathways peaked around 48 to 72 hr following the injury. By 30 days after the injury, no cell death pathways were observed. Proteins that mapped to cell survival pathways were represented at all eight time points following mTBI. The numbers of proteins that mapped to the pathway were similar at all time points. The inflammation pathway displayed the largest number of altered proteins following mTBI with what may be a biphasic response peaking at 24 hr (eight proteins) and 7 days (six proteins) following injury (*Figure 4*). The other context-dependent pathways did not show time-dependent changes in protein numbers during the 30-day period.

The time-dependent changes in the eight most frequently changed proteins observed in the cortical antibody arrays are shown in *Figure 5A–H*. The eight most frequently altered proteins were FASL, VEGF-B, CXCL16, Fractalkine, GAS-1, MIG, IL-1 ra and IL-17E. *Figure 5A–H* shows the individual animal protein levels and the mean and S.E.M. for each protein and time point. All proteins showed a uniform decrease or increase compared to the sham protein levels, albeit with different

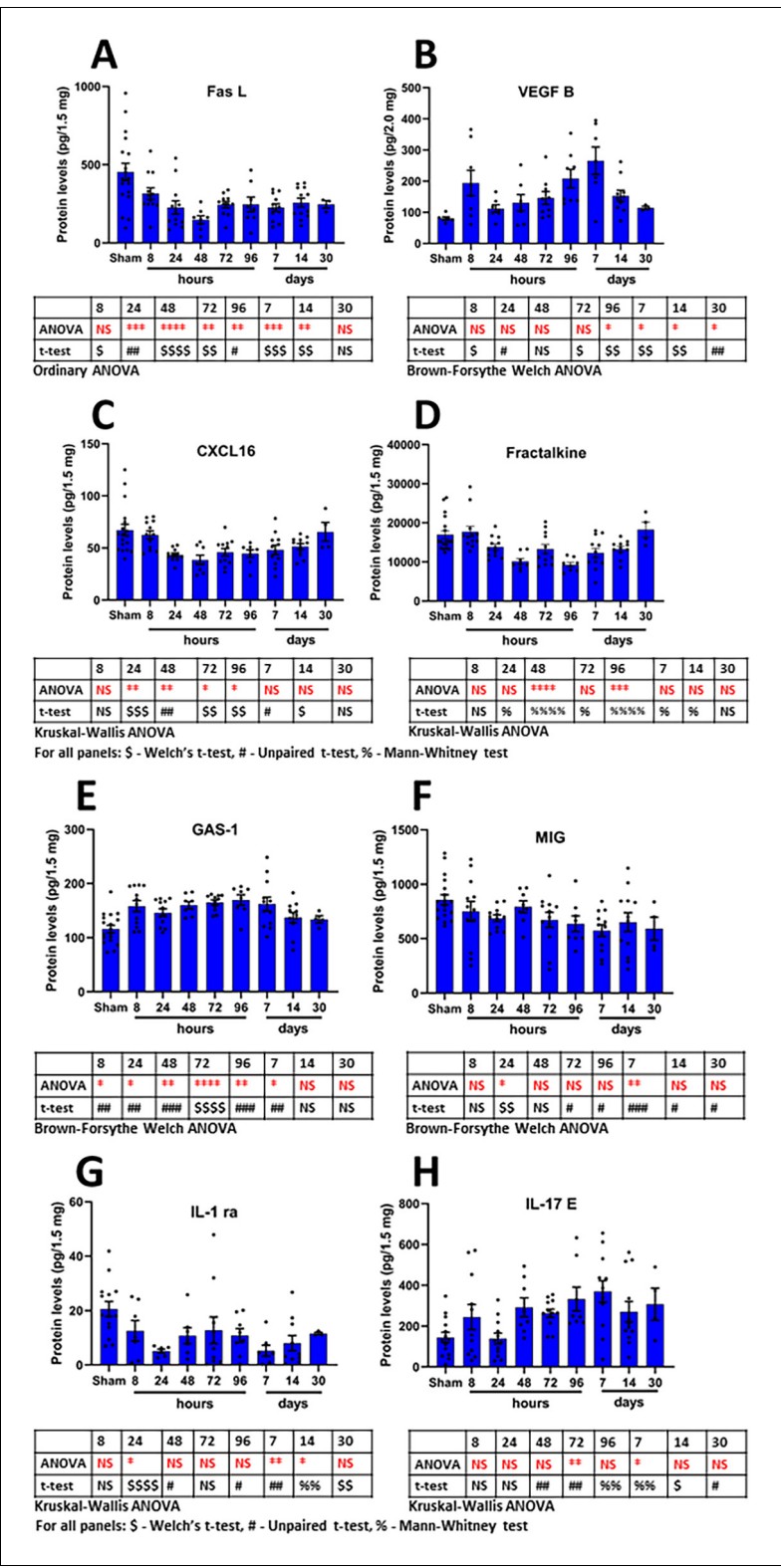

**Figure 5.** Time-dependent changes in the most frequently regulated cortical derived changed proteins induced by a single mTBI event over a 30 day period. Shown are the changes in proteins over time for the proteins FASL (A), VEGF-B (B), CXCL16 (C), Fractalkine (D), GAS-1 (E), MIG (F), IL-1 ra (G) and IL-17E (H). Provided are the mean ± S.E.M. data in bar graph form with the individual animal protein levels (filed circles). Below each component are the statistical markers and the type of ANOVA test that was performed, multiple comparisons

*Figure 5 continued on next page*

*Figure 5 continued*
were made between the mTBI time points and the sham group. Also, shown are the results of the Student's T-test, comparing the mTBI time point data with the sham data. Statistical analysis was performed with Prism Ver 8.31 (GraphPad). *Figure 5—source data 1*.
The online version of this article includes the following source data for figure 5:

**Source data 1.** Most frequently regulated cortical proteins.

degrees of variability. ANOVA and student's t-test significance markers are shown below each protein, additionally the proteins are listed in *Table 1A*.

There were 23 proteins identified to be significantly regulated at four or more times from the eight time points during the study. The proteins were relevant to several different pathways, protein classes and biological process classifications. The pathway with the most proteins identified from the group of 23 proteins was Inflammation mediated by cytokine and cytokine signaling (P00031, Fractalkine, MIP-1a and RANTES), and six additional pathways were identified: Alzheimer disease-presenilin (P00004, E-cadherin), Apoptosis signaling (P00006, FASL), CCKR signaling (P06959, E-cadherin), Cadherin signaling (P00012, E-cadherin), FAS signaling (P00020, FASL) and Wnt signaling (P00057, E-cadherin). Pathways have been defined elsewhere.

Four protein classes were identified by mapping the 23 proteins. The protein class with the largest number of proteins from the group of 23 was Signaling molecule (PC00207, FASL, MIP-1a, Galectin-1, CD40L, CXCL16, VEGF-B, RANTES, TCA-3 and MIG), then Receptor (PC00197, IL-1 r4 and MBL-2), cytoskeletal protein (PC00085, CRP) and Cell adhesion molecule (PC00069, Galectin-1).

Nine different biological processes were identified by mapping the proteins. Cellular process mapped the largest number of proteins from the group of 23, (GO:0009987, FASL, IL-1 r4, MIP-1a, Galectin-1, CD40L, MBL-2, CXCL16, MIG, VEGF-B, MadCam-1, E-cadherin, RANTES and TCA-3), followed by Response to stimulus (GO:0050896, FASL, MIP-1a, CD40L, MIG, VEGF-B, RANTES and TCA-3), Localization (GO:0051179, MIP-1a, CXCL16, MadCam-1, MIG, VEGF-B, RANTES and TCA-3) and Biological regulation (GO:0065007, MIP-1a, CXCL16, MadCam-1, VEGF-B, RANTES, Fractalkine and TCA-3). Several other biological processes were identified: Developmental process (GO:0032502, MBL-2), Biological adhesion (GO:0022610, MadCAM-1, E-cadherin), Multicellular organismal process (GO:0032501, MBL-2, VEGF-B), Cell proliferation (GO:0008283, VEGF-B) and Immune system process (GO:0002376, FASL, CD30L, CD40L, Fractalkine, CRP, MIG). All proteins significantly regulated at four or more of the eight time points are listed in *Table 1A*. The definitions of Protein Class and Biological Process are provided in *Table 1B*.

## NEV cargo proteins after mTBI

A summary of NEV particle concentration and average diameter data is shown in *Table 2*. There were no significant differences in NEV particle concentration and average diameter between NEV samples from sham animals or mTBI-challenged ones for any time point. Changes in NEV protein levels determined on the SOMAscan assay were small, this observation along with sample variability and low numbers of samples per time point made it challenging to obtain statistical significance comparing the mTBI samples with the sham samples. However, four of the common proteins between the different proteomic platforms showed differences between NEV derived from sham and mTBI-challenged animal samples assessed with the Kruskal-Wallis nonparametric ANOVA test. The proteins were CRP, p value = 0.0196; IL-6, p value = 0.0243; IL-13, p value = 0.0253; MIP-3b, p value = 0.0297, (*Figure 6A–D*). Performing post-tests and comparing TBI values with those of sham failed to demonstrate any significant differences in the proteins. The strongest positive correlation in protein measurements between the proteomic platforms was for the cytokine IL-10 (the correlation with matched cortical data was +0.56). The strongest negative correlation was for the protein TCK-1 (−0.40). The overall correlations ranged from +0.56 to −0.40 (*Supplementary file 2*, correlations between cortical tissue and NEV proteins).

## Discussion

The TBI animal model used in this study produces a mild diffuse form of injury with no focal lesion or disruption of the blood-brain barrier (BBB). Prior studies utilizing this model have identified evidence

**Table 1.** Mouse cortical tissue proteins significantly altered compared to sham tissue four or more times over the study period.
(A) Red font indicates increases in protein levels, blue font indicates reductions in protein levels, black font indicates that the proteins were observed to be both increased and reduced over the duration of the study. (B) Gene ontology definitions are as follows: *Pathways* are defined elsewhere. *Protein class* definitions - Signaling molecule (PC00207) - a molecule that is secreted by one cell and received as a signal by a receptor on another cell. Receptor (PC00197) - a protein or complex that spans the plasma membrane, that binds to a signal molecule in the extracellular space and transduces the signal to the cytoplasm. Cytoskeletal protein (PC00085) - a major constituent of the cytoskeleton found in the cytoplasm of eukaryotic cells. Cell adhesion molecule (PC00069) - a protein that mediates cell-to-cell adhesion. *Biological process* definitions - Cellular process (GO:0009987) - any process that is carried out at the cellular level, but not necessarily restricted to a single cell. Response to stimulus (GO:0050896) - any process that results in a change in state or activity of a cell or an organism as a result of a stimulus. Localization (GO:0051179) - any process in which a cell, a substance, or a cellular entity, such as a protein complex or organelle, is transported, tethered to or otherwise maintained in a specific location. Biological regulation (GO:0065007) - any process that modulates a measurable attribute of any biological process, quality or function. Biological adhesion (GO:0022610) - the attachment of a cell or organism to a substrate, another cell, or other organism. Biological adhesion includes intracellular attachment between membrane regions. Cell population proliferation (GO:0008283) - the multiplication or reproduction of cells, resulting in the expansion of a cell population. Developmental process (GO:0032502) - a biological process whose specific outcome is the progression of an integrated living unit: an anatomical structure (which may be a subcellular structure, cell, tissue, or organ), or organism over time from an initial condition to a later condition. Immune system process (GO:0002376) - any process involved in the development or functioning of the immune system, an organismal system for calibrated responses to potential internal or invasive threats. Multicellular organismal process (GO:0032501) - any biological process, occurring at the level of a multicellular organism, pertinent to its function.

**A – Identities of 23 significantly altered proteins**

| # of significant time points out of 8 | Protein |
|---|---|
| 7 | FASL, VEGF-B |
| 6 | CXCL16, Fractalkine, IL-1 ra, MIG, IL-17E, GAS-1 |
| 5 | IL-1 r4, MBL-2, Progranulin, CRP, E-cadherin |
| 4 | GITR, IL-20, CD30L, MIP-1a, RANTES, TCA-3, CD40L, Galectin-1, MadCam-1, Tweak R |

**B – Identities of significantly altered proteins mapped to different functional categories**

| Pathway | Protein |
|---|---|
| CCKR signaling map (P06959) | E-cadherin |
| Wnt signaling pathway (P00057) | E-cadherin |
| FAS signaling pathway (P00020) | FASL |
| Inflammation mediated by chemokine and cytokine signaling pathway (P00031) | Fractalkine, MIP-1a, RANTES |
| Apoptosis signaling pathway (P00006) | FASL |
| Alzheimer disease-presenilin pathway (P00004) | E-cadherin |
| Cadherin signaling pathway (P00012) | E-cadherin |
| **Protein Class** | **Protein** |
| Signaling molecule (PC00207) | FASL, MIP-1a, Galectin-1, CD40L, CXCL16, VEGF-B, RANTES, TCA-3, MIG |
| Receptor (PC00197) | IL-1 r4, MBL-2 |
| Cytoskeletal protein (PC00085) | CRP |
| Cell adhesion molecule (PC00069) | Galectin-1 |
| **Biological Process** | **Protein** |
| Cellular process (GO:0009987) | FASL, IL-1 r4, MIP-1a, Galectin-1, CD40L, MBL-2, CXCL16, MIG, VEGF-B, MadCam-1, E-cadherin, RANTES, TCA-3 |
| Localization (GO:0051179) | MIP-1a, CXCL16, VEGF-B, MadCam-1, RANTES, TCA-3, MIG |
| Biological regulation (GO:0065007) | MIP-1a, CXCL16, Fractalkine, VEGF-B, MadCam-1, RANTES, TCA-3 |
| Response to stimulus (GO:0050896) | FASL, MIP-1a, CD40L, MIG, VEGF-B, RANTES, TCA-3 |
| Developmental process (GO:0032502) | MBL-2 |
| Biological adhesion (GO:0022610) | MadCAM-1, E-cadherin |
| Multicellular organismal process (GO:0032501) | MBL-2, VEGF-B |

*Table 1 continued on next page*

| | |
|---|---|
| Cell proliferation (GO:0008283) | VEGF-B |
| Immune system process (GO:0002376) | FASL, CD30L, CD40L, Fractalkine, CRP, MIG |

of neuronal cell loss as early as 3 days post-injury, and neurobehavioral deficits that develop over time, typically starting from 7 days that remain present 30 days after the initial brain injury. We previously identified alterations in hippocampal expression of multiple genes, many of which suggest a strong involvement of inflammation and inflammatory pathways following the induction of TBI (*Tweedie et al., 2013a*; *Tweedie et al., 2016a*; *Tweedie et al., 2016b*; *Tweedie et al., 2013b*). In the present study, we investigated changes in brain tissue proteins related to inflammation and inflammatory processes over an extended time period, from 8 hr to 30 days post-mTBI, with eight different sampling time points following the induction of a single mTBI.

As no behavioral or immunohistochemical measures were evaluated in the mTBI animals over the time span of the current study, our only evidence of TBI-induced changes and trends in measures between the sham and mTBI animals can be made by comparisons of the mTBI and sham cortical and NEV proteins. Findings illustrate that the induction of the injury can induce marked, lasting changes in cytokine/chemokine proteins in mouse cortical tissue. We identify a subset of proteins that were observed to be significantly altered by mTBI in four or more of the eight sample time points (in all 23 proteins). Of these proteins a more focused smaller set of eight (*Table 1A*) were identified that were significantly altered in six to seven of the eight different time points evaluated following injury. It is possible that several of the TBI-altered proteins identified here may prove useful as biomarkers of TBI. Although these proteins warrant further evaluations to validate them as biomarkers of mTBI, this is beyond the scope of the current study. It is important to understand that following mTBI both cellular physiological and pathological processes are taking place, as evidenced by the mapping of several of the significantly altered cortical derived proteins to pathways associated with pro-survival - physiological, and cell death - pathological processes. In this study, three different proteins stood out as potential candidate proteins for development as drug targets. They were novel in the context of traumatic brain injury and displayed consistent increases or reductions

**Table 2.** Neuronally enriched EV particle counts and sizes for sham and mTBI time points obtained from mouse plasma.
SOMA EV Counts by treatment group, One-way Analysis of Variance (ANOVA), the p value is 0.3560, considered not significant. SOMA EV sizes by treatment group, One-way Analysis of Variance (ANOVA), the p value is 0.2198, considered not significant. *Table 2—source data 1*.

| | Sham | mTBI | mTBI | mTBI | mTBI | mTBI | mTBI |
|---|---|---|---|---|---|---|---|
| EV counts/ml | | hours | hours | hours | days | days | days |
| | | 8 | 24 | 72 | 7 | 14 | 30 |
| Mean | $6.83 \times 10^{10}$ | $6.54 \times 10^{10}$ | $5.41 \times 10^{10}$ | $4.93 \times 10^{10}$ | $3.86 \times 10^{10}$ | $4.42 \times 10^{10}$ | $8.43 \times 10^{10}$ |
| SD | $2.67 \times 10^{10}$ | $3.94 \times 10^{10}$ | $2.36 \times 10^{10}$ | $3.36 \times 10^{10}$ | $1.89 \times 10^{10}$ | $2.87 \times 10^{10}$ | $4.39 \times 10^{10}$ |
| SEM | $8.44 \times 10^{9}$ | $1.97 \times 10^{10}$ | $1.18 \times 10^{10}$ | $1.68 \times 10^{10}$ | $9.48 \times 10^{9}$ | $1.44 \times 10^{10}$ | $2.20 \times 10^{10}$ |
| N | 10 | 4 | 4 | 4 | 4 | 4 | 4 |
| P-value (vs. Sham) | | 0.8738 | 0.3753 | 0.2828 | 0.0677 | 0.1607 | 0.4138 |
| | | | | | | | |
| EV size (nm) | | | | | | | |
| Mean | 172 | 204 | 164 | 210 | 149 | 171 | 165 |
| SD | 38 | 39 | 16 | 58 | 38 | 19 | 24 |
| SEM | 12 | 20 | 8 | 28 | 19 | 9 | 12 |
| N | 10 | 4 | 4 | 4 | 4 | 4 | 4 |
| p-value (vs. Sham) | | 0.1919 | 0.7117 | 0.1710 | 0.3286 | 0.9536 | 0.7274 |

The online version of this article includes the following source data for Table 2:
**Source data 1.** EV particle.

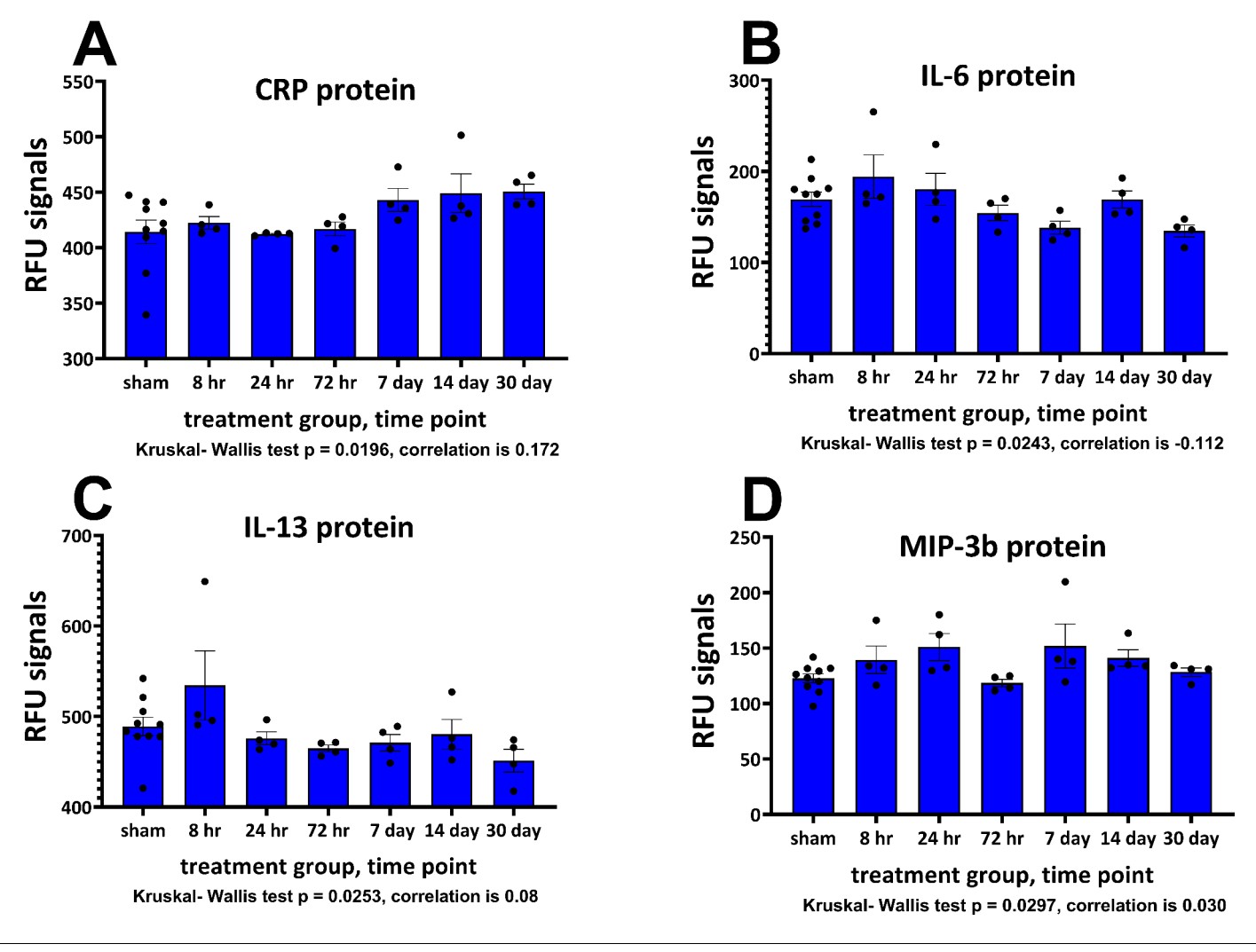

**Figure 6.** Plasma neuronal-enriched EV cargo proteins significantly altered by a single mTBI event over a 30-day period. The Relative Fluorescence Units (RFU) obtained from the SOMAscan assay for NEV proteins are shown. Provided are the mean ± S.E.M. data in bar graph form with the individual animal protein levels (filed circles). Due to the low numbers in the TBI groups, sample normality was tested with the Shapiro-Wilk test, all sham data passed normality, several of the TBI data points were not normally distributed and were subject to ANOVA by Kruskal-Wallis, p values are provided. Below each component are the correlation values comparing matched cortical and NEV sample data. Values are presented as the mean ± S.E.M. of n = 4–10 EV samples. *Figure 6—source data 1*.

The online version of this article includes the following source data for figure 6:

**Source data 1.** EV cargo protein.

during the study time frame. GAS-1 and VEGF-B were increased, and CXCL16 was reduced compared to sham levels. Of these three proteins GAS-1 and CXCL16 may be the best options for drug target validation, both proteins are known to modulate neuronal cell death involving glutamate excitotoxicity, a common feature of brain injuries (*Globus et al., 1995*; *Mellström et al., 2002*; *Rosito et al., 2012*; *Rosito et al., 2014*; *Sun et al., 2016*; *Wang et al., 2016a*; *Zarco et al., 2012*) and as such may make them ideal candidate drug targets for evaluation. While there are no direct reports of the actions of VEGF-B on the traumatized brain, it is tempting to speculate that the homeostatic and pro-neurogenesis properties of the protein would be ripe for exploration in TBI (*Nag et al., 2002*; *Sun et al., 2004*; *Sun et al., 2006*). After completion of studies aimed at defining if the observed protein changes were pathological or physiological compensatory in nature following mTBI, these proteins may make for good candidate drug targets. Pre-clinical studies manipulating

the proteins followed up with sensitive behavioral and immunohistchemical analysis will be vital in the validation of each protein as novel drug targets in mTBI.

Complex TBI-induced changes in cellular protein expressions and time-dependent changes in cortical proteins were observed in our study that mapped to common pathways related to pathological processes, namely apoptosis, Fas signaling and Alzheimer's disease. Commonly recurring cell survival - physiological pathways were related to angiogenesis, EGF receptor, PDFG, GDNF and Wnt signaling systems. Based on the mapping of the significantly changed proteins there appeared to be a balance between cell death and cell survival pathways as early as 8 hr and then up to 14 days after injury. It is interesting that by the last sample collection time point in our study, 30 days post-mTBI, the cell death associated pathways were not observed to be active, yet physiological and immunological pathways were observed. While we are not able to comment on the time between 14 days and 30 days, these data suggest that in this mild model of TBI a transition in the balance between proteins responsible for cell death and cell survival has occurred. Future work can be directed at identifying which proteins change between these times after a mTBI.

Comparing our data with data derived from brain RNA-Seq analysis (*Zhang et al., 2014*) showed that of the 23 proteins most frequently changed over time, that 10 of the proteins displayed preferential expression in microglia over astrocytes and neurons (VEGF-B, CXCL16, IL-1ra, progranulin, MIP-1a, RANTES, Galectin-1, TWEAK R, also MADCAM-1 and IL-1 r4 albeit with low expression levels in all cell types). While the proteins GAS-1, GITR and IL-17E showed preferential expression in astrocytes (GITR and IL-17E were of low expression levels in all three cell types). Fractalkine and FASL both showed a preferential expression in neurons over the other cells, yet FASL expression was low for brain tissue. Of the remaining proteins, eight proteins were observed to be expressed at very low levels in brain and showed no preferential expression between microglia, astrocytes and neurons (MIG, MBL-2, CRP, E-cadherin, IL-20, CD30 L, CD40 L and TCA-3). Based on these observations, one can suggest that the microglial cells are chiefly responsible for the generation of the proteins changed over time in the mouse cortical tissue post-mTBI and, as such, may make a good candidate for more focused drug target interventions. To establish a treatment regimen that targets microglia following a TBI of any level of severity, it is important to realize that the microglial cell possesses a dual character where it can be linked with both protective and damaging cellular responses. The dual nature of the microglial cell has been thought of as being either M1 or M2, a classification borrowed from that of peripheral macrophages (*Mills et al., 2000*), which are linked to pro-inflammatory or anti-inflammatory functions, respectively. Classically, it is believed that both forms of cells express distinct markers exclusively depending on the status of the cell. In more recent times, data relating to CNS microglia markers has provided evidence that a microglial cell may simultaneously express markers classically associated with M1 and M2 phenotypes. A study examining gene expressions and proteins following an open head controlled cortical impact (CCI) TBI (*Morganti et al., 2016*), provides such evidence of mixed populations of microglial cells. Morganti and co-workers describe the expression of numerous proteins that were formerly associated with one form or the other (M1/M2) in the same cells. Several of the proteins they describe were observed in our antibody array data, for example those associated with M1: MIG (CXCL9), CXCL16, and those associated with M2: Fractalkine (CX3CL1), IL-17E also known as IL-25, and TGFβ1, to name a few. These observations further suggest that even in mild models of TBI, such as our model, that brain tissue microglia present with mixed phenotypes following injury.

Two proteins were significantly changed compared to sham at seven of the eight times after mTBI; specifically, Fas Ligand (FASL) and VEGF-B. FASL is a protein bound to the cell membrane or cleaved from it to generate a soluble protein that has a role in apoptosis. Here, we show that cortical tissue levels of FASL were reduced as early as 8 hr following mTBI, protein levels reached their lowest point at 48 hr with a partial recovery that appeared to be maintained until 14 days following mTBI. Our findings differ somewhat to those of others who have investigated FAS signaling following TBI. Beer and co-workers have shown evidence of increased immunostaining of FAS/FASL in an open head, moderate CCI model of TBI in rats (*Beer et al., 2000*), a finding in contrast to our observations from our mild closed head model of TBI. Topcu and co-workers described increased FAS/FASL immunostaining in rat brain following a closed head weight drop at 24 hr after injury (*Topcu et al., 2013*). Zhang and co-workers demonstrated changes in FAS receptor protein levels, but no changes in FASL protein in postmortem human severe TBI compared to non-TBI control tissues (*Zhang et al., 2003*). These conflicting reports may be due to differences in TBI severity and

mechanisms of injury induction. It is noteworthy that Ziebell and co-workers reported that mice with mutated FAS receptor and disrupted FAS signaling subjected to open scalp, closed head weight drop (a model of TBI similar to ours) had better neurological outcomes and pathological changes when compared to wild-type animals (*Ziebell et al., 2011*). This suggests that attenuating FAS signaling may be beneficial to the traumatized brain. It is possible that the reduction in tissue FASL, seen herein, is a homeostatic compensatory response to mTBI. This hypothesis may be addressed by further studies employing pharmacological intervention(s) with proven efficacy in our mild model of TBI.

The growth factor vascular endothelial growth factor-B (VEGF-B) displayed a more complex pattern of expression. Its protein level was significantly increased at 8 and 24 hr following TBI; however, although also elevated versus sham animals at 48 hr after injury, levels did not attain statistical significance at this time. Notably, from 72 hr up to and including 30 days following mTBI, VEGF-B protein levels were again significantly elevated. The roles of VEGF-B in physiology and disease is nicely described in review by Bry and co-workers (*Bry et al., 2014*). They indicate that VEGF-B has anti-apoptotic effects and is beneficial in cardiovascular disease and neuroprotective in the setting of cerebral ischemia and animal models of Amyotrophic Lateral Sclerosis and Parkinson's disease. They also suggest that the described benefits of VEGF-B in CNS are meditated by the receptor VEGFR1 located on neurons and that the expression of VEGF-B may be regulated at the tissue level by mitochondrial metabolic processes. Somewhat in confliction with these observations are those from studies by Rothhammer and co-workers (*Rothhammer et al., 2018*). They describe a complex relationship between microglial derived TGF-alpha and VEGF-B on the pathological role of astrocytes in a murine model of multiple sclerosis, with VEGF-B exacerbating the disease pathology. Interestingly, Rothhammer and co-workers provided evidence that dietary metabolites of tryptophan were able to regulate microglial activation and the generation of VEGF-B, suggesting a role of the gut microbiome in the regulation of brain inflammation. Presently there are no reports documenting altered levels of VEGF-B in the setting of TBI. The protein is constitutively expressed in the vascular endothelium, and it may have a beneficial role in stabilizing or maintaining BBB function. The genetic deletion of VEGF-B is reported to exacerbate neurological disease, while elevations in brain vascular levels of VEGF-B may provide a compensatory restorative action aimed at stabilizing TBI-induced BBB damage. Additionally, the protein has been shown to display pro-neurogenesis effects in rodent brain (*Nag et al., 2002*; *Sun et al., 2004*; *Sun et al., 2006*). In the light of this information, it is likely that VEGF-B elevations observed in this study are a homeostatic compensatory response aimed at minimizing TBI-induced disturbances in BBB function and stabilizing the brain microenvironment. A reduction in BBB integrity, subtle changes in blood flow, and vasogenic edema are common following TBI and can drive the activation of proteases, initiation of inflammatory pathways and production of reactive oxygen species to further cause tissue damage (*Alluri et al., 2015*). Time-dependent studies in rats subjected to CCI injury demonstrated a biphasic reduction in BBB integrity, peaking initially at 4 to 6 hr after injury and then at 3 days (*Başkaya et al., 1997*), which largely coincides with the VEGF-B biphasic changes in our mTBI model.

Six proteins were significantly different at six of the eight time points after mTBI compared to sham; specifically, CXCL16; Fractalkine, IL-1 ra and MIG were lower, IL-17E and GAS-1 were higher after mTBI compared to sham. This may be the first study to report alterations in cortical protein levels of the chemokine CXCL16 following mTBI. CXCL16 was shown to be consistently reduced starting 24 hr following mTBI and remained so for up to 14 days following injury. Reports of the involvement of CXCL16 in brain injury are limited; however, this protein has been reported to have potential neuropathological or neuroprotective roles in neurological disorders. For example, in the setting of multiple sclerosis CXCL16 has been linked with neuropathology (*Hendrickx et al., 2013*); yet it has also been shown to possess neuroprotective properties in the setting of stroke and in vitro excitotoxicity studies. In this regard, CXCL16 can attenuate the neurotoxic effects of glutamate on neurons (*Rosito et al., 2012*; *Rosito et al., 2014*). Excitotoxicity caused by glutamate is a common observation in most forms of brain insult, including TBI (*Globus et al., 1995*). The decline in CXCL16 seen in our study suggests a loss of protection from excitotoxicity as a causative factor for mTBI-induced neurodegeneration.

Fractalkine (CX3CL1) is a chemokine that can exist as a soluble or a membrane bound protein that binds exclusively to one receptor (CXC3R1). The protein has been shown to have a role in the regulation of the activity of microglial cells. In the present study, reductions in fractalkine cortical

levels, compared to sham levels, were first observed at 24 hr after TBI and followed a bi-phasic response peaking at 48 hr, and then at 96 hr after TBI. Reduced protein levels were maintained for up to 14 days following injury. There is conflicting evidence regarding the physiological role of fractalkine in brain, some studies suggest that it tends to push microglial cells towards a more activated phenotype (*Cardona et al., 2006*), while others to a more neuroprotective one (*Mizuno et al., 2003*; *Gaetani et al., 2013*). The consequences of fractalkine signaling on microglial phenotype appear to be directed, in part, by the nature of the CNS injury and whether it is an acute or chronic insult. Fractalkine has been shown to be neuroprotective in vitro, against challenges with glutamate or N-methyl-D-aspartate in hippocampal neurons (*Lauro et al., 2015*; *Limatola et al., 2005*). *Cardona et al., 2006* demonstrated that deletion of the fractalkine receptor from microglia exacerbated neurotoxicity when microglia were activated by various stimuli. Febinger and co-workers (*Febinger et al., 2015*) illustrated that fractalkine signaling played a role in TBI pathology following a CCI injury. In that study, they used a fractalkine receptor knock-out animal and showed that a lack of fractalkine signaling led to fewer TBI-related functional deficits (apoptosis and neuronal cell death) compared to wild-type mice, at early times after injury (less than 15 days), while at later times (approximately 30 days following injury) the transgenic animals showed more deficits compared to wild-type animals. This finding suggests that microglial phenotypes are strongly sensitive to time-dependent alterations in fractalkine signaling, shifting microglial phenotypes from a more anti-inflammatory (neuroprotective) state to a more pro-inflammatory (neurotoxic) state at later times. This interpretation is further supported by a study by Zanier and co-workers using a similar, yet slightly more severe model of CCI brain injury than that used in the Febinger study (*Zanier et al., 2016*). Clausen and co-workers illustrated significant elevations of fractalkine in cerebrospinal fluid after a fluid percussive injury in rats (*Clausen et al., 2019*). Subsequent to stroke, fractalkine mRNA levels and immunohistochemical reactive signals for fractalkine were observed to be reduced in the infarct area, but increased in tissues surrounding the infarct core (*Tarozzo et al., 2002*). In contrast to our observations, Rancan and co-workers observed no changes in brain tissue levels of fractalkine in a mouse closed head injury model somewhat similar in nature to our model (*Rancan et al., 2004*).

The cytokine IL-1 receptor antagonist (IL-1 ra) was markedly reduced compared to sham levels at 24 hr after TBI. Changes in levels of this cytokine were more complex than those observed for other proteins; there appeared to be a bi-phasic response. IL-1 ra protein levels were significantly reduced compared to sham levels even on day 30 after injury. In the setting of TBI there are conflicting reports on the utility of employing IL-1 ra as a potential treatment of TBI, some suggest that treatment can push microglial cells to a more activated pro-inflammatory phenotype, with others indicating protective effects in the CNS following TBI (*Toulmond and Rothwell, 1995*; *Knoblach and Faden, 2000*; *Helmy et al., 2014*; *Helmy et al., 2016*; *Semple et al., 2017*; *Sun et al., 2017*; *Tehranian et al., 2002*). In a mouse model of mTBI, not dissimilar to ours, Tehranian and co-workers observed that mice overexpressing brain IL-1 ra fared better than wild-type animals in overall neurological assessments, and possessed reduced levels of pro-inflammatory cytokines following TBI. Our observations with mTBI show that IL-1 ra levels are reduced compared to sham protein levels. Our findings indirectly add further credence to the premise that augmentation of IL-1 ra, in the setting of mTBI, may push the brain micro-environment toward a beneficial state that may provide a useful treatment modality in the clinic. Shiozaki and co-works reported elevated CSF concentrations of IL-1 ra after TBI in humans that depended on its severity, with higher concentrations observed after more severe TBIs (*Shiozaki et al., 2005*).

This may be the first report of TBI causing a sustained reduction of the chemokine MIG (also known as CXCL9). MIG/CXCL9 is thought to play a beneficial role in precursor neuronal cell development, survival and integration into existing neural networks (*Turbic et al., 2011*). A reduction of MIG may reflect the commonly observed altered neurogenesis after TBI (*Ngwenya and Danzer, 2018*). Wang and co-workers reported that altered states of neurogenesis following TBI were, in part, dependent on the nature and severity of the injury; reporting severity-dependent changes in neural stem cell proliferation, and the generation of immature and then mature neurons following graded CCI injuries (*Wang et al., 2016b*).

Interestingly, the signaling protein growth arrest-specific 1 (GAS-1) is highly expressed in the brain and has been associated with neuronal cell death (*Mellström et al., 2002*; *Sun et al., 2016*; *Wang et al., 2016a*; *Zarco et al., 2012*). Wang and co-workers illustrated that GAS-1 protein levels in vitro, were increased by a challenge with glutamate (a hallmark feature of TBI) and that elevated

levels of GAS-1 protein may interfere with the protective effects of glial cell neurotrophic factor (GDNF, *Wang et al., 2016a*). Our studies revealed that GAS-1 protein levels were elevated as early as 8 hr following mTBI, peaking at 72 hr and remaining elevated for up to 7 days after injury. In this light, the effects of mTBI on GAS-1 protein levels and the functional properties of this protein underscore it as a prospective novel drug treatment target and also as a monitoring biomarker for TBI.

In the present study, cortical protein levels of IL-17E (also known as IL-25), were robustly elevated by 48 hr after mTBI, remaining so even at 30 days after injury. Reports of the involvement of the proinflammatory protein IL-17E in the setting of TBI are absent. However, IL-17E/IL-25 has been shown to be elevated in plasma in a neuroinflammatory disorder, neuromyelitis optica spectrum disorder (*Zhang et al., 2018*). There is evidence that IL-25 is generated in brain by capillary endothelial cells and that this cytokine may, like VEGF-B, play a beneficial homeostatic role in the brain by maintaining the BBB integrity and, in addition, may drive microglial cells towards an anti-inflammatory phenotype (*Sonobe et al., 2016*; *Maiorino et al., 2013*). Elevations in IL-25 have been observed in cortical and hippocampal tissues of patients with epilepsy (*Kan et al., 2012*).

Whereas the changes in proteins described above may provide interesting drug targets to either mitigate or monitor TBI, it is likely that differences in their onset time, duration of regulation, as well as dynamic range in their level of change, may suggest different contexts of use as biomarkers. For example, due to the early pattern of mTBI-induced changes in the proteins FASL, CXCL16 and GAS-1, these may have utility as biomarkers for early TBI. IL-17E (IL-25) presented with a more delayed regulation after TBI, and may thus provide insight into later stages of TBI development and possible responses to pharmacological interventions in relation to injury recovery. Similarly, VEGF-B was significantly increased at 8 and 24 hr, but not at 48 hr, but then again at all time points from 72 hr onwards, and may, like 1 L-17E, be useful for following changes in brain tissue responses to injury or pharmacological interventions at mid to late time points. In contrast, IL-1 ra and fractalkine both showed marked changes in protein levels; however, the patterns were complex bi-phasic responses. Such time-dependent changes in protein levels can still provide utility as biomarkers of disease progression or drug response, but should probably not be interpreted individually, but as part of a panel of other biomarkers to provide insight into the nature of the biological processes that are active at different times (*Parolo et al., 2018*).

It is noteworthy that several of the proteins identified to be changed by a single mTBI over a 30 day period have been described to modulate neuronal cell death mediated by excitotoxic stimulation, such as by glutamate. This is significant, since excessive release of glutamate following brain injury is considered a hallmark feature of brain trauma (*Globus et al., 1995*). CXCL16 was reduced in our model, and has been shown to be involved in cellular communications between neurons and glial cells, and to possess neuroprotective properties that can attenuate the neurotoxic effects of glutamate in vitro and excitotoxic stimulation in vivo (*Rosito et al., 2012*; *Rosito et al., 2014*). Similarly, fractalkine, which was shown to be reduced after mTBI in our study, has been shown to be neuroprotective, in vitro, against challenge with glutamate or N-methyl-D-aspartate (*Lauro et al., 2015*; *Limatola et al., 2005*). The protein GAS-1 was elevated in our model of mTBI; it is interesting that Wang and co-workers illustrated that GAS-1 protein levels were up-regulated in vitro by challenge with glutamate and that elevated levels of GAS-1 protein may interfere with the protective effects of glial cell neurotrophic factor (GDNF, *Wang et al., 2016a*). These observations give added weight to the importance of glutamate excitotoxicity following brain trauma, and further provide hopeful drug targets for evaluation in additional TBI models and in the clinic.

Changes in NEV cargo proteins over time were relatively small compared to changes in cortex, and only moderate correlations were observed. This finding is, perhaps, not surprising as we assessed EVs enriched for neuronal origin and it is widely considered that microglia and astrocytes generate the largest amounts of cytokines and chemokines of brain cell types; nevertheless, neurons also produce them to some extent (*Gadient and Otten, 1994*; *Ringheim et al., 1995*). It is, however, noteworthy that one of the significantly regulated NEV proteins did show some similarity between the two sample types. The protein, c-reactive protein (CRP), is regulated by inflammation and is present in neurons (*Yasojima et al., 2000*). It displayed significant increases in cortex over time, with non-significant elevations in later time points in CNS EV-derived samples. There are numerous reports linking neurotrauma and TBI with elevated CRP and poor patient outcomes (*Bengzon et al., 2003*; *Hergenroeder et al., 2008*; *Gullo et al., 2011*; *Ishikawa et al., 1999*; *Sharma et al., 2017*). For the most part, reported CRP measurements were derived from patient

plasma/serum and not from EVs. It is possible that evaluation of CRP could prove to be useful singly or within a panel of markers of mTBI, and represents one of several candidate biomarkers meriting further evaluation.

In the present study, it is not possible to say to what extent the changes in proteins observed in the cortical arrays are explicitly due to reductions or increases in protein synthesis or if the changes are due to a loss or increase in cell numbers in the injured cortex. It is, however, worth considering that our prior work using this mild model of TBI with a 30 g weight has provided evidence of diffuse neuronal cell degeneration observed at 72 hr following mTBI, additionally we have observed losses of neurons at 7 and 30 days following mTBI (*Bader et al., 2019a*, *Bader et al., 2019b*; *Benady et al., 2018*; *Lecca et al., 2019*; *Rubovitch et al., 2015*). As such, we are not able to exclude the possibility that the reductions in the proteins observed in the cortical arrays are due to a loss of cells in the animal brains and not solely due to a reduction in expression of the proteins. However, it is important to note that numerous proteins were shown to be increased compared to sham tissue levels, suggesting that there are complex interactions between multiple cell types in the brain tissue microenvironment.

The work of others in the field of TBI has nicely shown that with more severe levels of injuries the greater are the functional deficits and cellular markers of pathology. This has been shown to be the case in such models as moderate fluid percussion injury (FPI) and moderate CCI TBI (*Shojo et al., 2010*; *Yu et al., 2009*). Yu et al., show that different grades of CCI injury result in severity-dependent losses in brain tissue and impaired measures of cognition and motor function. Shojo and co-workers showed time-dependent changes in gene expressions relating to inflammation that peaked shortly after PFI, that preceded the development of hallmark features of apoptosis at later times. Compared to our model of TBI, these are more severe models requiring the generation of craniotomies, with the subsequent disruption of the BBB. It is interesting to note that in our model of mild TBI, Tashlykov and co-workers showed that increasing the severity of injury by increasing the weights used to induce the TBI (drop height of 80 cm with weights from 5 to 30 g) resulted in a greater response regarding the induction of markers of apoptosis and neurodegeneration (*Tashlykov et al., 2007*; *Tashlykov et al., 2009*). Thus, it is possible that the use of different weights in our model of TBI may alter the time course and magnitude of the changes in proteins observed in our antibody arrays, this would require additional studies to verify this hypothesis. Also, it may be worth considering the effects of skull fractures in weight drop models as described by Zvejniece and co-workers. They describe work in an open head, close skull weight drop model (drop height of 8 cm with a weight of 90 g) examining the effects of different weight cone diameters (2 or 5 mm) on inducing TBIs associated with frequent skull fractures in mice (*Zvejniece et al., 2020*). Their observations describe worsened functional outcomes and more pronounced inflammatory responses for animals that received the weight drops which resulted in the occurrence of skull fractures. The authors suggest that under reported skull fractures may, in part, explain the frequently observed large variations seen in behavioral and biochemical outcome markers present in weight drop models of TBI. Our model does not use stereotactic instrumentation and, as a result, this leads to variability where the weight connects with the animal skull. Hence, we and others have observed variability in functional and biochemical measures in our model, which have been described previously by Israelsson and co-workers (*Israelsson et al., 2009*) and observed in this study (see *Figure 5*) and are in accord with human TBI.

Potential weaknesses of this study are related to a lack of behavioral assessments in the same animals from which tissue samples were derived. However, based on our prior studies in separate cohorts of alike mice, we demonstrated that mTBI, as used herein, resulted in consistent cognitive impairments involving both spatial (Y-maze) and visual (novel object recognition) tasks when evaluated at either 7 or 30 days after injury (*Zohar et al., 2003*; *Milman et al., 2005*; *Tweedie et al., 2013a*; *Tweedie et al., 2016a*). A further potential weakness may be related to a lack of corresponding immunohistochemical data; however, again our prior studies in alike cohorts of mTBI-challenged mice have consistently demonstrated the development of diffuse neuronal loss, neuroinflammation and astrogliosis (*Bader et al., 2019a*; *Lecca et al., 2019*; *Baratz et al., 2015*). As the animals used in this study were not perfused with buffer prior to euthanasia, there will have been a small volume of blood present in the dissected cortical tissues (*Kaliss and Pressman, 1950*; *Greig et al., 1988*). As such, we are not able to exclude the possibility that a small component of the array data may be derived from the mouse blood. Technical limitations of the SOMAscan assay

primarily reside in the fact that the detection system is not able to distinguish between post-translationally modified and non-modified proteins, and thus measurements reflect total protein abundances. Also, the assay is more qualitative than quantitative in nature. The strength of the assay lies in the capacity to assess a substantial number of target proteins at one time from a single sample.

The overall aim of the present study was to identify novel biomarkers related to inflammatory proteins following a single mild TBI in mice that may translate to human clinical TBI. A frequently used, powerful tool available to aid in biomarker discovery is mass-spectrometry (MS). A significant feature of MS is that the technique may provide information on thousands of proteins in one sample providing an unbiased approach to biomarker discovery. Interestingly, MS can be semi-quantitative or quantitative which involves the incorporation of specific proteins of interest to act as internal standards, an intervention that makes this aspect of MS more of a biased detection method rather than an unbiased approach. As all techniques have limitations, possible drawbacks of MS detection lie in sample preparation which can be time consuming and often requires several processing steps that may lead to over manipulation of the samples. Often high abundant background proteins may mask low abundance proteins of interest, which may then introduce artifacts in measuring low abundance proteins of relevance to the biology of the samples. Additionally, one sample can only be processed with MS at one time, and thus when working with large numbers of samples it would be time consuming to use MS technology. We chose to study the effects of a single mild TBI on inflammatory proteins over an extended time period in mouse cortical tissues, based on prior observations made in our animal model of mTBI (*Israelsson et al., 2009*; *Tweedie et al., 2016b*). With our focus on inflammatory proteins, we chose to exclude commonly investigated markers of TBI like NFL, UCH-L1 and GFP as, while they are worthy proteins of interest in TBI, they are not inflammatory in nature. Current available multiplex ELISA-based technologies targeting inflammatory proteins are limited to 10 or so targets. To investigate larger numbers of inflammatory proteins would require large quantities of samples. Hence, due to the limited availability of sample material from mouse cortical tissue, we elected to use a panel of quantitative antibody arrays. Potential limitations of antibody arrays may be related to the number of capture antibody spots attached to the array chip. Our arrays had 50 targets per chip, requiring the use of several different chips to investigate 200 proteins. It is interesting to note that a main utility of antibody arrays lies in the combination of high affinity, highly specific antibody pairs that allows for the sensitive detection of low and high abundance target proteins in complex sample lysate mixtures without the need to remove higher concentration background proteins. While this biased approach of biomarker investigation has inherent limitations, namely that we will miss out on proteins not present on the arrays, we were successful in our approach as we were able to demonstrate lasting changes in eight proteins, 5 of which are novel in the setting of TBI (FASL, VEGF-B, CXCL16, MIG, GAS-1, IL-17E, IL-1 ra and Fractalkine). While both MS and antibody array technologies have advantages and disadvantages, and both require subsequent target validation by alternative methods, it is likely that the combination of the unbiased MS and biased antibody array methods can complement each other to be used together to advance the discovery of novel proteins related to disease diagnosis, progression and potential drug target engagement.

The necessity of extracting tissue at specific time points underlies the difficulties in designing a study to both measure time-dependent changes in brain proteins to determine how they may regulate brain neuropathology and also assess animal behavior. These difficulties may be overcome by the use of non-invasive CNS sampling methods, such as plasma derived neuronal extracellular vesicle protein cargo assessments. While these combined proteomic approaches were undertaken to provide insight into new possible biomarkers for mTBI, and the identification of functional pathways derived from significantly changed cortical proteins, additional studies are clearly required with a more focused aim to understand how pharmacological interventions may impact the levels of these candidate biomarkers in both rodent brain and NEVs from humans and animals, in parallel with cognitive, behavioral and neuropathological assessments. Correlations between the common proteins observed on the cortical arrays and the SOMAscan assay platform were weak. This is most likely caused by the small changes seen in the NEV proteins and the low animal numbers. More focused studies are required to investigate correlations between NEV cargo and cortical proteins following mTBI. Only with such information will the full utility of NEVs as non-invasive sources of possible biomarkers of TBI be confirmed, and only at this point we will be able to make any recommendations for candidate human markers of TBI and drug target engagement.

## Conclusion

Our findings present evidence of robust, long-lasting changes in cortical protein derived cytokines and chemokines following a single mTBI in mouse. Several of the more frequently changed proteins were novel in the setting of mTBI, namely CXCL16, GAS-1 and VEGF-B, and warrant further investigation as possible drug targets and biomarkers of TBI that may translate to the clinic. Additional studies are required to fully understand the potential role of NEVs and other CNS cell type EVs as non-invasive sources of CNS biomarkers related to mild TBI. These studies will be vital in defining the utility of EV proteins as brain injury severity markers and treatment response markers.

## Materials and methods

Mouse mTBI studies were conducted at the Intramural Research Program of the National Institute on Aging, Baltimore, MD, USA. Specifically, male CD-1 mice, 6–8 weeks old (weighing approximately 30 g) were housed five to a cage under a 12 hr light/dark cycle (constant temperature 22°C) with food (Purina rodent chow) and water ad libitum. Experimental animal protocols were approved by the Animal Care and Use Committee of the Intramural Research Program, National Institute on Aging (438-TGB-2022) and were in compliance with the guidelines for animal experimentation of the National Research Council (Committee for the Update of the Guide for the Care and Use of Laboratory Animals, 2011) and the National Institutes of Health (DHEW publication 85–23, revised, 1995). A minimal number of mice were used for the study and well-being evaluations were applied to ensure a lack of potential suffering.

Experimental mTBI was induced using the concussive, closed-scalp head trauma device described previously (*Zohar et al., 2003*; *Milman et al., 2005*; *Tashlykov et al., 2007*; *Tweedie et al., 2007*). Mice were fully anesthetized by inhalation anesthesia in a closed isoflurane vaporizer chamber. After full anesthesia had been achieved, each animal was placed under a metal tube device with the opening positioned directly over the animal's head just anterior to the left ear (to provide mild injury to the left cerebral hemisphere). Each mouse was held in a manner such that the force of impact to the skull generated anterior-lateral movements without any rotational movements, analogous to those that occur during closed head injury in automobile accidents. The injury was induced by dropping a 1 cm diameter blunted flat cylindrical metal weight (30 g), inside a metal tube device (inner diameter 13 mm) from a height of 80 cm. This injury is analogous to two humans of the same weight clashing heads. Sham (control) animals were subjected to the same procedure, but without the weight being dropped. After the induction of the injury or sham procedure, mice were placed back in their home cage to allow for recovery from the anesthesia. Mice were randomly assigned to receive either mTBI or sham injury, and those in the former group were then randomly assigned to be euthanized at different post-injury times (*Table 3*). Due to the observations that even in this mild model of TBI, TBI-induced alterations in brain cells occur in both the ipsilateral and contralateral hemispheres (*Tashlykov et al., 2007*; *Tashlykov et al., 2009*), we have included absolute sham operated control

**Table 3.** Mouse sample treatment/time points following mTBI for cerebral cortex (CTX) and plasma-sampled, neuronally enriched EV samples.

| Treatment/Time Point | CTX (Cytokine/Chemokine Arrays, n) | NEV (SOMAscan, n) |
| --- | --- | --- |
| sham | 18 | 10 |
| mTBI 8 hr | 12 | 4 |
| mTBI 24 hr | 12 | 4 |
| mTBI 48 hr | 8 | ND |
| mTBI 72 hr | 12 | 4 |
| mTBI 96 hr | 8 | ND |
| mTBI 7 Days | 12 | 4 |
| mTBI 14 Days | 12 | 4 |
| mTBI 30 Days | 4 | 4 |

ND refers to not determined.

animals for comparison with mTBI animal protein measurements. In the design of this study, we chose to include one group of sham animals to be used to compare the effects of mTBI over time rather than including a sham group for each mTBI time point. In brief, we studied a total of 18 sham animals; 12 animals euthanized at each of 8, 24 and 72 hr, 7 and 14 days post-mTBI; eight animals euthanized at each of 48 and 96 hr post-mTBI and four animals euthanized at 30 days post-mTBI evaluation. Selection of animal numbers was based on our prior studies (*Israelsson et al., 2009*; *Tweedie et al., 2007*; *Tweedie et al., 2016a*; *Tweedie et al., 2016b*). Whole blood was collected from animals that were fully anesthetized and euthanized by decapitation, blood was placed in heparinized tubes. Plasma was prepared from the blood and NEVs were isolated from the plasma. All animals were euthanized in the same way to avoid introducing sampling variations.

## RayBiotech cytokine arrays

Cerebral cortical tissue (specifically, left parietal cortex (the side of injury)) was obtained from sham and mTBI animals (ipsilateral to the mTBI) and immediately frozen to −80˚C. Tissue supernatants underwent quantitative assessment of cytokine/chemokine proteins performed by RayBiotech Inc (Norcross, Georgia) utilizing the Quantibody Mouse Cytokine Array 4000 assay kit (*Shi et al., 2016a*), in five quantitative arrays (QAM-CYT-4,–5, −6,–7, −8 arrays). The arrays are similar to traditional sandwich-based ELISAs only capture antibodies for a number of protein targets are attached to a glass slide in an array format which allows for the multiplex detection of more proteins in one sample at one time. QAM-CYT-4,–5 and −6 were performed with protein loading of 1.5 mg/ml; QAM-CYT-7 and −8 were performed with protein loading of 1.0 and 2.0 mg/ml, respectively.

To understand the functional roles of the significantly changed proteins in the cortical tissue, we took the identities of the mouse proteins/genes and applied them to a web-based tool to examine functional classifications of the altered proteins - Protein ANalysis THrough Evolutionary Relationships (PANTHER; http://pantherdb.org, *Mi et al., 2019*). We analyzed the proteins that were significantly changed in four and more of the eight time points used in our study and identified pathways, biological process and protein class.

## EV isolation and subsequent characterization

Heparinized whole blood was collected from a subset of animals at the time of euthanasia. The blood was centrifuged within 1 hr to prepare plasma (10,000 *g* for 5 mins at 4˚C), which was stored immediately at −80˚C. The study groups utilized for NEV isolations were as follows: sham, n = 10; 8, 24 and 72 hr post-mTBI, each one, n = 4; 7, 14 and 30 days post-mTBI, each one, n = 4 (*Table 3*). Total nanoparticles were precipitated from plasma using a commercially available kit, Exoquick TM (Systems Biosciences), followed by enrichment for EVs of neuronal origin by immunoadsorption with an antibody selective for L1 CAM (L1 Cell Adhesion Molecule) (*Shi et al., 2016b*; *Goetzl et al., 2016*). The concentration and average diameter of NEVs were assessed by Nanoparticle Tracking Analysis; each NEV sample was diluted 1:100 in PBS, then the diluted samples were processed using Nanosight NS500 (*Goetzl et al., 2018*). Subsequently, the remaining NEVs were lysed using M-PER (Pierce Thermo) and two freeze-thaw cycles, and total protein concentrations were determined using the BCA assay (Pierce Thermo).

## SOMAscan assay

The SOMAscan assay platform uses a technology where chemically modified single stranded oligomers with specific protein binding capabilities are coupled to a capture surface in wells on an assay plate. Each well on the plate has approximately 1300 specific oligomers and hence protein targets. Oligo-protein binding fluorescent signals are used to generate qualitative assessments in the changes of NEV proteins in the NEV lysates. The SOMAscan assay (SomaLogic Inc, Boulder, Colorado), which has the capacity to measure over 1300 proteins in a single sample (*Kiddle et al., 2015*; *Voyle et al., 2015*; *Sattlecker et al., 2014*), was performed on NEV lysates by the Center for Human Immunology, National Institutes of Health, Bethesda, MD, USA. NEV lysates were applied to the SOMAscan assay using equal protein loading conditions (60 µg/ml), and samples were assayed using the SOMAscan Assay Kit 1.3 k (Cells and Tissue). SOMAscan assay data on protein abundance were expressed as relative fluorescence units (RFUs) and were converted into percentage differences

compared to sham. We focused our analysis of NEV proteins to a subset that overlapped with significantly regulated proteins in the cortical tissue (94 proteins).

## Statistical Analysis

Cortical and NEV derived protein measurements were subject to statistical analysis using GraphPad Prism (8.3.1.). The protein measurements underwent several statistical assessments: (1) outliers were identified and removed from further analysis, (2) data normality was assessed, (3) student's unpaired t-tests and (4) ANOVAs were performed. Cortical tissue measurements were subjected to the Grubb's test to identify significant outlier data points, any outliers were excluded from further analysis (*Grubbs, 1950*). Data normality was assessed by use of the Kolmogorov-Smirnov test or the Shapiro-Wilk test. To determine significant differences between mTBI and sham protein levels we used the following assessments. If data were normally distributed with equal variance, an ordinary student's unpaired t-test or an ordinary ANOVA followed by Bonferroni's multiple comparisons test was used. If data were normally distributed, but they did not have equal variance an unpaired students t-test with Welch's correction and a Brown-Forsythe and Welch ANOVA was performed with a Dunnett's T3 multiple comparisons test. If data were not normally distributed non-parametric statistical tests were used; a Mann-Whitney student t-test was used and a Kruskal-Wallis ANOVA test followed by a Dunn's multiple comparisons test. Data are presented as the mean ± standard error of the mean (SEM) of n observations, n refers to the number of samples (mice).

## Acknowledgements

This study was supported in part by (1/2) the Intramural Research Program of the NIH, National institute on Aging, (3) the Ari and Regine Aprijaskis Fund at Tel-Aviv University and the Dr. Miriam and Sheldon G. Adelson Chair for the Biology of Addictive Diseases in Tel-Aviv University, Tel-Aviv, Israel, and (4) National Institutes of Health R56 AG057028. The authors wish to acknowledge RayBiotech Inc, (Norcross, Georgia) for performing the Chemokine/Cytokine protein arrays and the Center for Human Immunology, National Institutes of Health, Bethesda, Maryland, USA, for performing the SOMA Logic SOMAscan assay.

## Additional information

### Funding

| Funder | Grant reference number | Author |
| --- | --- | --- |
| National Institutes of Health | AG000944 | Nigel H Greig |
| National Institutes of Health | R56 AG057028 | Barry J Hoffer |
| Ari and Regine Aprijaskis Fund | | Chaim G Pick |
| Dr. Miriam and Sheldon G. Adelson Chair for the Biology of Addictive Diseases | | Chaim G Pick |

The funders had no role in study design, data collection and interpretation, or the decision to submit the work for publication.

### Author contributions

David Tweedie, Conceptualization, Formal analysis, Investigation, Methodology, Writing - original draft, Writing - review and editing; Hanuma Kumar Karnati, Formal analysis, Investigation; Roger Mullins, Formal analysis, Methodology, Writing - original draft, Writing - review and editing; Chaim G Pick, Conceptualization, Writing - original draft; Barry J Hoffer, Conceptualization, Writing - original draft, Writing - review and editing; Edward J Goetzl, Dimitrios Kapogiannis, Writing - original draft, Writing - review and editing; Nigel H Greig, Conceptualization, Resources, Funding acquisition, Writing - original draft, Project administration, Writing - review and editing

## Author ORCIDs
David Tweedie (iD) https://orcid.org/0000-0002-8446-4544
Nigel H Greig (iD) http://orcid.org/0000-0002-3032-1468

## Ethics

Animal experimentation: All animal studies were conducted at the Intramural Research Program of the National Institute on Aging, Baltimore, MD, USA. Experimental animal protocols were approved by the Animal Care and Use Committee of the Intramural Research Program, National Institute on Aging (438-TGB-2022) and were in compliance with the guidelines for animal experimentation of the National Research Council (Committee for the Update of the Guide for the Care and Use of Laboratory Animals, 2011) and the National Institutes of Health (DHEW publication 85-23, revised, 1995).

## Decision letter and Author response

Decision letter https://doi.org/10.7554/eLife.55827.sa1
Author response https://doi.org/10.7554/eLife.55827.sa2

# Additional files

### Supplementary files

• Source data 1. Cortical tissue data. This file contains the raw data obtained from the Antibody Array studies undertaken with the five different arrays. Provided are the different treatment groups, the protein names, the upper and lower limits of the assay detection and the individual data points as pg. The assay proteins loading conditions have been described in the methods section. The sample identity is provided. This file contains all of the original data related to *Figures 1–5*, describing the cortical tissue protein changes over time.

• Source data 2. NEVs data. This file contains the raw data in relative fluorescent units for each of the proteins measured in the neuronal extracellular vesicles. Provided are target full name, target, uniprot, entrez gene id, entrez gene symbol. The sample identity is provided. These data relate to the protein measurements from the plasma derived neuronal extracellular vesicles that were screened on the SOMASCAN assay, *Figure 6*.

• Supplementary file 1. Proteins significantly regulated by a single mTBI event over time. Provided are lists of up-regulated and down-regulated proteins, also the mean, SEM and number of values for each time point. Data are expressed a percentage increase from sham and complement the source data files.

• Supplementary file 2. Correlations between cortical and NEV proteins. Provided is a list proteins that were significantly changed in the cortical proteins that were common between the cortical array and SOMAscan assay.

• Transparent reporting form

### Data availability

All data generated by US NIH funded research is available to the public, the data generated in this study is available to the public. The data used to generate the figures in the manuscript are provided as source data files.

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
