## [Decision Letter]

**Acceptance summary:**

The study by Tweedie et al. provides an elaborate temporal profile of the changes to protein expression in the rodent brain following a mild traumatic brain injury (mTBI). In this descriptive study, a screen of 200 inflammatory proteins identified novel and confirmed changes to protein expression in the cerebral cortex following a single mTBI. These data provide new insight into the biological cascades that follow mTBI and offer several putative biomarkers for targeting and evaluating therapeutics.

**Decision letter after peer review:**

Thank you for submitting your article "Time-dependent cytokine and chemokine changes in mouse cerebral cortex following a mild traumatic brain injury" for consideration by *eLife*. Your article has been reviewed by four peer reviewers, one of whom is a member of our Board of Reviewing Editors, and the evaluation has been overseen by Tadatsugu Taniguchi as the Senior Editor. The following individual involved in review of your submission has agreed to reveal their identity: Cesar Borlongan (Reviewer #4).

The reviewers have discussed the reviews with one another and the Reviewing Editor has drafted this decision to help you prepare a revised submission. In recognition of the fact that revisions may take longer than the two months we typically allow, until the research enterprise restarts in full, we will give authors as much time as they need to submit revised manuscripts.

Summary:

Tweedie et al. conduct a screen for proteins that temporally change in the mouse brain following mild traumatic brain injury. The unpredictable nature of traumatic brain injury (TBI) contributes to the difficulty in developing effective therapeutics. The temporal changes in protein expression following TBI may provide biomarkers to both identify disease severity and evaluate therapeutic efficacy. Using a weight-drop model of TBI, the authors quantify protein levels in the impacted mouse cortex at different time points post-TBI and examine proteins in neuron-derived exosomes extracted from blood. The authors identify 23 proteins that were altered at specific time points, and via gene ontology pathway mapping implicated these proteins as being associated with inflammation, apoptosis, Alzheimer disease and angiogenesis, providing insights into the potential use of these proteins are disease markers.

Overall, the study is a protein screen for putative biomarkers in TBI. The study appears technically sound with some questions from the reviewers to be addressed. One of the major issues is the lack of a validated finding from this screen e.g. a candidate biomarker is identified, validated and shown to have translational relevancy. Given the limitation that a preselected panel of proteins is being examined, this is not an unbiased approach and follow up validation of FasL or CRP, for example, would support that the approach used is identifying potential biomarkers. In lieu of validation, a more definitive conclusion and summary is needed for follow up studies which is reflected in Essential Revisions based on the comments from the four reviewers. A second major concern is the presentation of the data should more easily reflect temporal changes in putative biomarkers of TBI. Suggestions are given below.

Essential revisions:

1) Independent validation of the array data using an independent assay is not provided and noted to be beyond scope of paper by the authors; however, a validation experiment would lend further support to top candidate(s) that are most likely to track with injury for multiple times after injury (e.g. FasL). In lieu of validation, a succinct summary of what the authors view as strongest biomarkers for further pursuit based on their findings would be useful to the field. For example, the authors identify proteins that are altered over the majority of time points (>4/8 time points post-TBI) in Table 1B. Is it possible to sort or provide "priority score" to which process may be most relevant based on proteins? This would help guide reader to the most potentially useful "biomarker" for TBI and at what time point. Perhaps a summary figure/table of what the authors deem top targets for further validation if further validation is indicated as being beyond scope of the paper. Overall, presenting more of the data as protein changes over time and discussing different trends/patterns that are observed would provide more meaningful interpretation of the data. Furthermore, discussing how those proteins trends might reflect/influence stages of TBI pathophysiology would be insightful.

For example, for Figure 1B, consider histogram showing relative abundance change so that it is easier to see the temporal effect of for each pathway. Same for Figures 2B, 3B and 4. A common key could be used and plot histogram same order as key (or sort histogram by highest to lowest change then put key in the same relative order). Also consider supplemental figure showing histogram of pathway changes for all time points (could be broken into 3-4 graphs on a single page) or just proteins from Table 1. This would be useful to see how they are changing over time rather than only broken into early and late.

2) The proteins identified were linked to inflammation, apoptosis, and Alzheimer disease, which are generally associated with cell death signaling pathways. However, some of the proteins were also linked to angiogenesis, which is normally considered as a cell survival pathway. Accordingly, it will be important for the authors to briefly clarify that some proteins that were altered post-TBI may be cell death, while others are cell survival proteins. Similarly, briefly in the Abstract, but more detailed in the Discussion, the authors may also wish to discuss how such cell death and cell survival associated proteins can be targeted for therapeutic development.

3) Because a main theme of this study is "time-dependent" protein expression post-TBI, clarifying any overt or subtle changes in cell death and cell survival proteins over time post-TBI will be of interest. One would likely assume that in the very early phase of TBI that there will endogenous protection, thus the angiogenic proteins may be more upregulated than the pro-inflammatory, apoptotic and AD-relevant cell death signals. Vice versa, during the late phase of TBI (several days or weeks), then the protein level expression may be reversed, now with pro-cell death protein signals more prominent than cell survival proteins. A discussion along this topic may be interesting.

4) Relevant to item 3, the authors have published extensively in neuroinflammation, and thus aware of the double-edge sword phenomenon of inflammation, in that inflammation may be beneficial to some extent in the early stages of TBI, but detrimental in the late phases of TBI. Additionally, that the M1/M2 polarization may not be that distinct (e.g., see papers by Rosi Brain Circ. 2016; Morganti et al., 2016; Marcet et al., Neuroimmunol Neuroinflamm. 2017). Some insights to this phenomenon based on the expression of inflammatory proteins over time post-TBI will be informative.

5) What are effects on different cell types in the cortex? For example, if microglia ar3e enriched after TBI in the region, wouldn't it be logical that microglia proteins are enriched, and this can be also seen as increase in many of the proteins that are classified as apoptotic (since many of them are expressed in microglia). On the other hand, frakalkine is mostly secreted by neurons and acting on microglia/macrophages. Would it be possible to do similar analysis as can be done for transcriptomic data e.g. gene set enrichment analysis (GSEA, Zhang et al., 2014, J. Neurosci.), that would indicate what is the most affected cell type in each time points?

6) It is important to state the limitations of the array approach used here in that it represents 200 preselected proteins.

a) What was the basis for this selection?

b) Why were some of the more standard proteins/potential biomarkers discussed within the context of TBI, for example GFAP, NFL, and UCH-L1 not included for reference?

c) While all experimental approaches have their limitations, what are the inherent limitations of the antibody-based approach used here as opposed to mass-spectrometry based methods?

7) The authors end the Results section by mentioning that IL-10 and TCK-1 have the strongest positive and negative correlations between protein arrays and NEV analysis. Were they significantly changed or was magnitude to small? Why not highlight this finding more? At minimal, it warrants mention in Discussion.

8) CRP was downregulated for extended periods in the EVs starting from day 7. Does any of the EV findings correlate with poor recovery and could this be used as biomarker for TBI?

9) Since the authors used the mild model of TBI, some insights as to whether the moderate and severe TBI model will accelerate or exacerbate the protein expression will be of interest. Despite the mild TBI, did the authors detect any correlations between animals with more accelerated or exacerbated protein expression when the severity of their injury is worse than those with very mild TBI? A short discussion of their mild TBI with previous papers on moderate TBI may be interesting (see for example Shojo et al., 2010; Shojo et al., Cell Transplantation 2017).

10) Did the authors discern any interesting trends among the injured animals? Such trends should be added to Discussion.

11) It would be more informative if authors gave the measurements for the proteins from individual animals rather than just the average for each time point.

[Editors' note: further revisions were suggested prior to acceptance, as described below.]

Thank you for resubmitting your article "Time-dependent cytokine and chemokine changes in mouse cerebral cortex following a mild traumatic brain injury" for consideration by *eLife*. Your revised article has been reviewed by three peer reviewers, one of whom is a member of our Board of Reviewing Editors, and the evaluation has been overseen by Tadatsugu Taniguchi as the Senior Editor. The following individual involved in review of your submission has agreed to reveal their identity: Cesar Borlongan.

The reviewers have discussed the reviews with one another and the Reviewing Editor has drafted this decision to help you prepare a revised submission.

Summary:

The revised manuscript identifies top 3 candidate biomarkers and the overall data presentation aids the reader in understanding the temporal changes in protein expression following mTBI. The reviewer's concerns were largely addressed. A concern over statistical analysis has been raised and requires attention before consideration for publication as it may impact interpretation. As noted in the first reviews, a primary concern was the lack of independent validation of the protein biomarkers which were identified by an array screen. In the revision, no validation was performed. As such, this is a descriptive screening study identifying three potential biomarkers of the protein changes following a mild traumatic brain injury.

Revisions:

The statistical analyses including the use of different ANOVA and t-tests should be clearly stated. A more detailed description of the statistical analyses is needed as the Materials and methods section appears discrepant with figures.

---

## [Author Response]

Essential revisions:1) Independent validation of the array data using an independent assay is not provided and noted to be beyond scope of paper by the authors; however, a validation experiment would lend further support to top candidate(s) that are most likely to track with injury for multiple times after injury (e.g. FasL). In lieu of validation, a succinct summary of what the authors view as strongest biomarkers for further pursuit based on their findings would be useful to the field. For example, the authors identify proteins that are altered over the majority of time points (>4/8 time points post-TBI) in Table 1B. Is it possible to sort or provide "priority score" to which process may be most relevant based on proteins? This would help guide reader to the most potentially useful "biomarker" for TBI and at what time point. Perhaps a summary figure/table of what the authors deem top targets for further validation if further validation is indicated as being beyond scope of the paper. Overall, presenting more of the data as protein changes over time and discussing different trends/patterns that are observed would provide more meaningful interpretation of the data. Furthermore, discussing how those proteins trends might reflect/influence stages of TBI pathophysiology would be insightful.For example, for Figure 1B, consider histogram showing relative abundance change so that it is easier to see the temporal effect of for each pathway. Same for Figures 2B, 3B and 4. A common key could be used and plot histogram same order as key (or sort histogram by highest to lowest change then put key in the same relative order). Also consider supplemental figure showing histogram of pathway changes for all time points (could be broken into 3-4 graphs on a single page) or just proteins from Table 1. This would be useful to see how they are changing over time rather than only broken into early and late.

We thank the reviewers for the useful comments and suggestions. This revision point also touches on essential point # 2. As such in the discussion we have taken the list of the 8 most frequently changed proteins over the 30 day time period and we have narrowed down the potential biomarker targets to 3 proteins (GAS-1, CXCL16 and VEGF-B). We believe that these proteins have the strongest potential for biomarkers most worthy of further investigation. The proteins were selected due to the facts that they were identified to be novel in the setting of TBI and uniformly regulated over the duration of the study.

We agree with the reviewer that the use of pie charts (as seen in the previous version of Figures 1B, 2B and 3B (and Figure 4)) did not provide as much helpful data as we hoped relating to the pathways mapped from the proteins over time. We have therefore removed the pie charts from the figures, and in our revised manuscript we have described the changes in the pathways and the proteins that mapped to the pathways in the text for each Results section. In an attempt to show time-dependent changes in proteins/pathways in a graphical manner, we have generated a new set of graphs showing the numbers of proteins that mapped to the pathways (y axis) with each time point (x axis). The graphs were broken down into pathological – cell death, physiological – cell survival and pathways that were contextually dependent on the nature of the proteins that mapped to the pathways (i.e. pro- or anti-inflammatory cytokines/interleukins), Figure 4.

Furthermore, we agree that the presentation of the data over time provides a better insight to the possible utility of the proteins as biomarkers. As many of the proteins were only significantly changed at 1 or 2 time points, we decided to focus on the 8 most frequently changed proteins. We have generated a new Figure 5 showing the time-dependent changes in the 8 most frequently changed proteins. Provided on the figure are the individual data points reflecting each animal and the mean ± standard error of the mean of the data. Below each of the 8 components of the figure are tables indicating the statistical markers for each time point when assessed by ANOVA (also the type of ANOVA is indicated) and by the Student’s t-test. For all analyses, the different mTBI time points were compared to the sham values.

2) The proteins identified were linked to inflammation, apoptosis, and Alzheimer disease, which are generally associated with cell death signaling pathways. However, some of the proteins were also linked to angiogenesis, which is normally considered as a cell survival pathway. Accordingly, it will be important for the authors to briefly clarify that some proteins that were altered post-TBI may be cell death, while others are cell survival proteins. Similarly, briefly in the Abstract, but more detailed in the Discussion, the authors may also wish to discuss how such cell death and cell survival associated proteins can be targeted for therapeutic development.

We thank the reviewer for raising the points about clarifying the observations that some of the proteins mapped to both pathological and physiological cellular processes. We have clarified this observation in the Abstract, Results and Discussion sections. We have discussed likely drug targets based on the specific proteins identified during our study, this is in the Discussion section.

“It is important to understand that following mTBI both cellular physiological and pathological processes are taking place, as evidenced by the mapping of several of the significantly altered cortical derived proteins to pathways associated with pro-survival – physiological, and cell death – pathological processes. […] Pre-clinical studies manipulating the proteins followed up with sensitive behavioral and immunohistchemical analysis will be vital in the validation of each protein as novel drug targets in mTBI.”

3) Because a main theme of this study is "time-dependent" protein expression post-TBI, clarifying any overt or subtle changes in cell death and cell survival proteins over time post-TBI will be of interest. One would likely assume that in the very early phase of TBI that there will endogenous protection, thus the angiogenic proteins may be more upregulated than the pro-inflammatory, apoptotic and AD-relevant cell death signals. Vice versa, during the late phase of TBI (several days or weeks), then the protein level expression may be reversed, now with pro-cell death protein signals more prominent than cell survival proteins. A discussion along this topic may be interesting.

We thank the reviewer for providing us the opportunity to discuss the time-dependent changes in pathological and physiological pathways and the proteins that mapped to these pathways. We have added the following paragraph to the discussion describing the time dependent changes in pathways observed in our study.

“Complex TBI-induced changes in cellular protein expressions and time-dependent changes in cortical proteins were observed in our study that mapped to common pathways related to pathological processes, namely apoptosis, Fas signaling and Alzheimer’s disease. […] Future work can be directed at identifying which proteins change between these times after a mTBI.”

4) Relevant to item 3, the authors have published extensively in neuroinflammation, and thus aware of the double-edge sword phenomenon of inflammation, in that inflammation may be beneficial to some extent in the early stages of TBI, but detrimental in the late phases of TBI. Additionally, that the M1/M2 polarization may not be that distinct (e.g., see papers by Rosi Brain Circ. 2016; Morganti et al., 2016; Marcet et al., Neuroimmunol Neuroinflamm. 2017). Some insights to this phenomenon based on the expression of inflammatory proteins over time post-TBI will be informative.

We thank the reviewer for this thoughtful suggestion, we have added a paragraph on the phenomena to the Discussion section.

“Based on these observations one can suggest that the microglial cells are chiefly responsible for the generation of the proteins changed over time in the mouse cortical tissue post-mTBI and, as such, may provide a good candidate for more focused drug target interventions. […] These observations further suggest that even in mild models of TBI, such as our model, that brain tissue microglia present with mixed phenotypes, following injury.”

5) What are effects on different cell types in the cortex? For example, if microglia ar3e enriched after TBI in the region, wouldn't it be logical that microglia proteins are enriched, and this can be also seen as increase in many of the proteins that are classified as apoptotic (since many of them are expressed in microglia). On the other hand, frakalkine is mostly secreted by neurons and acting on microglia/macrophages. Would it be possible to do similar analysis as can be done for transcriptomic data e.g. gene set enrichment analysis (GSEA, Zhang et al., 2014, J. Neurosci.), that would indicate what is the most affected cell type in each time points?

This is a very interesting question, the excellent article the reviewer refers to extracted very pure and specific cell types from rodent brain and then studied them with RNA-seq and cDNA gene array analysis. We worked with heterogeneous samples on a series of antibody arrays. As many of the proteins were changed at a single time point, we believed it may be more helpful to investigate the 23 proteins that were regulated in 4 or more times over the 30 day duration of the study. We took these proteins and we screened them through the Barres lab Brain RNA-Seq database (https://www.brainrnaseq.org/). The database provides information on the expression of gene transcripts in several different cell types found in brain. We focused on neurons, microglia and astrocytes. Using this tool we were able to identify which of those three cell types were most likely responsible for the generation of the 23 most frequently changed proteins observed on the antibody arrays.

“Comparing our data with data derived from brain RNA-Seq analysis (Zhang et al., 2014) showed that of the 23 proteins most frequently changed over time, that 10 of the proteins displayed preferential expression in microglia over astrocytes and neurons (VEGF-B, CXCL16, IL-1ra, progranulin, MIP-1a, RANTES, Galectin-1, TWEAK R, also MADCAM-1 and IL-1 r4 albeit with low expression levels in all cell types). […] Based on these observations one can suggest that the microglial cells are chiefly responsible for the generation of the proteins changed over time in the mouse cortical tissue post-mTBI and, as such, may make a good candidate for more focused drug target interventions.”

6) It is important to state the limitations of the array approach used here in that it represents 200 preselected proteins.a) What was the basis for this selection?b) Why were some of the more standard proteins/potential biomarkers discussed within the context of TBI, for example GFAP, NFL, and UCH-L1 not included for reference?c) While all experimental approaches have their limitations, what are the inherent limitations of the antibody-based approach used here as opposed to mass-spectrometry based methods?

We thank the reviewer for raising these important points relating to the choice of using antibody arrays over other biomarker discovery methods such as mass-spectrometry. We chose to study inflammation related proteins based on prior observations made in our animal model of mild traumatic brain injury where we illustrated that a single mild TBI event was able to induce changes in many genes related to inflammatory gene ontologies and molecular pathways, from very early times up to and including 14 days following the induction of the injury (Israelsson et al., 2009; Tweedie et al., 2016b). After a survey of commercially available quantitative single-plex and multi-plex ELISAs assay systems focused on inflammatory proteins we decided to use a commercial, quantitative antibody array approach. We fully understood that this method would introduce a strong bias in our biomarker discovery efforts towards inflammation related targets and that the data would be restricted to only 200 proteins. Even with such limitations we believed that the choice of these arrays met our research requirements and mitigated these limitations. Ultimately, our goal was to identify novel inflammatory proteins that may act as markers of mild TBI, a task where we believe that we were successful. We have been able to show lasting changes in 5 novel proteins, CXCL16, MIG, GAS-1, VEGF B and IL-17E induced by a single mild TBI over a 30 day follow up period. We accept the reviewers’ criticism that we failed to incorporate commonly investigated markers of TBI like NFL, UCH-L1 and GFP into our study. Our rational for this is, while GFAP is derived from brain immune cells, it is a structural protein and not a cytokine/chemokine. NFL and UCH-L1 are well documented neuronal proteins and not entirely novel to TBI or inflammation related, hence we excluded them from our analysis in this study. We have added the following text to the discussion describing our rational for selecting the antibody arrays and for the exclusion of the more standard TBI markers.

“The overall aim of the present study was to identify novel biomarkers related to inflammatory proteins following a single mild TBI in mice that may translate to human clinical TBI. […] While both MS and antibody array technologies have advantages and disadvantages, and both require subsequent target validation by alternative methods, it is likely that the combination of the unbiased MS and biased antibody array methods can complement each other to be used together to advance the discovery of novel proteins related to disease diagnosis, progression and potential drug target engagement.”

7) The authors end the Results section by mentioning that IL-10 and TCK-1 have the strongest positive and negative correlations between protein arrays and NEV analysis. Were they significantly changed or was magnitude to small? Why not highlight this finding more? At minimal, it warrants mention in Discussion.

We thank the reviewer for pointing out our omission in referring to the correlations between the cortical and NEV levels of IL-10 and TCK-1. The changes in the levels of the proteins measured in the NEV samples on the SomaLogic proteomic platform were small, with fairly large variations between samples. This and the added difficulty with low numbers of animals made it difficult to see any robust statistical differences between the NEV mTBI and the sham samples. Changes in both IL-10 and TCK-1 failed to achieve any statistical significance in NEV samples. We have added the following text to the Results section and the Discussion.

Results:

“Changes in NEV protein levels determined on the SOMAscan assay were small, this observation along with sample variability and low numbers of samples per time point made it challenging to obtain statistical significance comparing the mTBI samples with the sham samples. However, 4 of the common proteins between the different proteomic platforms showed differences between NEV derived sham and mTBI-challenged animal samples assessed with the Kruskal-Wallis nonparametric ANOVA test. The proteins were CRP, p value = 0.0196; IL-6, p value = 0.0243; IL-13, p value = 0.0253; MIP-3b, p value = 0.0297, (Figure 6 A-D). Performing post-tests comparing TBI values with sham valued failed to demonstrate any significant differences in the proteins.”

Discussion:

“Correlations between the common proteins observed on the cortical arrays and the SOMAscan assay platform were weak. This is most likely caused by the small changes seen in the NEV proteins and the low animal numbers. More focused studies are required to try to investigate correlations between NEV cargo and cortical proteins following mTBI, only then will the full utility of NEVs as non-invasive sources of possible biomarkers of TBI be confirmed, at which point will we be able to make any recommendations for candidate human markers of TBI and drug target engagement.”

8) CRP was downregulated for extended periods in the EVs starting from day 7. Does any of the EV findings correlate with poor recovery and could this be used as biomarker for TBI?

We thank the reviewer for the question about the utility of CRP as a possible NEV biomarker for TBI, on re-evaluation of the NEV data over time we determined that, while there were significant Kruskal-Wallis values, there were in fact no statistically significant differences between the mTBI groups and the sham group. Based on these observations we do not believe that plasma derive NEV CRP would be a useful biomarker for TBI.

9) Since the authors used the mild model of TBI, some insights as to whether the moderate and severe TBI model will accelerate or exacerbate the protein expression will be of interest. Despite the mild TBI, did the authors detect any correlations between animals with more accelerated or exacerbated protein expression when the severity of their injury is worse than those with very mild TBI? A short discussion of their mild TBI with previous papers on moderate TBI may be interesting (see for example Shojo et al., 2010; Shojo et al., Cell Transplantation 2017).

We thank the reviewer for this suggestion, in the present study all the mTBI animals were subjected to the same level of “weight drop” intensity, i.e. a 30g weight and no metrics other than cortical and NEV protein measurements were made. Thus, it is a little difficult to discuss in a definitive manner the possible effects of more intense injuries on changes in temporal and the magnitude of changes in cortical proteins over time. However, it is of interest to note that in the same animal model as that used in the present study Tashlykov and co-workers (Tashlykov et al., 2007 and 2009), showed that increased injury severity, by use of weights ranging from 5 g up to 30 g, also increased the levels of markers of degenerating neurons and pro-apoptotic proteins following TBI. This was observed to be the case more so in cortex, with more muted effects in hippocampus. Based on those observations one could theorize that for this specific model of TBI the use of different weights to induce different severities of injury may well induce altered levels, and possibly altered time-dependent changes in the proteins identified on the antibody arrays. Taking into consideration these studies and the work of others who have investigated different levels and models of injury on TBI severity on the subsequent CNS pathology, we have added the following paragraph to the discussion (Shojo et al., 2010, Yu et al., 2009, Zvejniece et al., 2020).

“The work of others in the field of TBI has nicely shown that with more severe levels of injuries the greater are the functional deficits and cellular markers of pathology. […] We and others have observed variability in functional and biochemical measures in our model, which have been described previously by Israelsson and co-workers (Israelsson et al., 2009) and observed in this study (see Figure 5).”

10) Did the authors discern any interesting trends among the injured animals? Such trends should be added to Discussion.

We thank the reviewer for this question, we did not incorporate any behavioral or immunohistochemical measurements in the study and as a consequence, we did not see any pathological or physiological trends in the injured animals (however, our prior studies on the same animal model has demonstrated consistent cognitive impairments in TBI challenged animals from early time points to 30 days) (see for example Bader et al., 2020 and 2019).

We have added a sentence to the discussion to reflect that we did not observe any trends in altered behaviors or other markers in the mTBI animals.

“As no behavioral or immunohistochemical measures were evaluated in the mTBI animals over the time span of the study, our only evidence of TBI-induced changes and trends in measures between the sham and mTBI animals can be made by comparisons of the mTBI and sham cortical and NEV proteins.”

11) It would be more informative if authors gave the measurements for the proteins from individual animals rather than just the average for each time point.

We thank the reviewer for this suggestion, we have generated 2 new graphs, 1 relating to the 8 proteins significantly changed in 6 and more of the 8 different time points (new Figure 5) and the other graph shows the 4 significant NEV derived proteins (new Figure 6). The graphs show the individual animal values and the mean and standard error of the mean for the given proteins.

[Editors' note: further revisions were suggested prior to acceptance, as described below.]

Revisions:The statistical analyses including the use of different ANOVA and t-tests should be clearly stated. A more detailed description of the statistical analyses is needed as the Materials and methods section appears discrepant with figures.

We thank the reviewers for the recommendation to clarify the statistical analysis. We have moved some of the text around and we have generated a new “Statistical Analysis” section in the Materials and methods section relating to the statistical analysis used in the study.